# Multi-Criteria Decision Making of Contractor Selection in Mass Rapid Transit Station Development Using Bayesian Fuzzy Prospect Model

**Min-Yuan Cheng** [1], **Shu-Hua Yeh** [1,*] **and Woei-Chyi Chang** [2]

[1]  Department of Civil and Construction Engineering, National Taiwan University of Science and Technology, Taipei 10672, Taiwan; myc@mail.ntust.edu.tw

[2]  Department of Civil Engineering, Construction Engineering and Management Division, National Taiwan University, Taipei 10617, Taiwan; woeichyichang@gmail.com

*   Correspondence: swyeh520520@gmail.com or d10205007@mail.ntust.edu.tw; Tel.: +886-2-2733-0004

**Abstract:** In Taiwan, the most advantageous tender in governmental procurement is the selection of a general contractor based on a score or ranking evaluated by a committee. Due to personal, subjective preferences, the contractor selection of committee members may be different, causing cognitive difference between the results of the members' selection and the preliminary opinions provided by the working group. Integrated, multi-criteria decision making techniques, combined with preference relation, Bayesian, fuzzy utility, and prospect theories are used to assess factors weighing up the duration/cost/quality, probability of external information, and utility function system. The paper proposes a Bayesian fuzzy prospect model for group decision making, based on probability and utility multiplied relation, and taking the sustainable development factors into consideration. This study aims to provide committees with an objective model to select the best contractor for public construction projects. The results of this study can avoid the lowest bidder being selected; besides, the score gap of contractor selection can be increased, and the difference between the top three contractors' scores can be decreased as well. In addition to proposing an innovative decision-making system of contractor selection and an index weight-assessing system for sustainable development, this model will be widely applied and sustainably updated for other cases.

**Keywords:** sustainability; multi-criteria decision making (MCDM); contractor selection; preference relation theory (PRT); Bayes' theorem (BT); fuzzy utility (FU); prospect theory (PT); risk preference

## 1. Introduction

Multiple criteria decision making (MCDM) is considered a complex decision-making tool involving both quantitative and qualitative factors. In recent years, several MCDM techniques and approaches have been suggested to choose the optimal probable options [1]. Such applications have been widely investigated in both the theory and practice of MCDM [2]. In the present study, a MCDM method was developed as follows. First, factors pertaining to duration, cost, and quality were determined. These factors are influential factors in MCDM [3,4]. Subsequently, the fuzzy preference relation (FPR) was adopted to construct a paired decision-making matrix of preferences [5]. This method enables decision makers to express preferences regarding a set of alternatives using the least number of judgments; the method also makes it unnecessary to examine the consistency of the decision-making process [6]. A fuzzy analytical hierarchy process (FAHP) method was used to prioritize the identified risks [7–9], and Fuzzy-TOPSIS achieved through the application of order preference by Similarity to Ideal Solution and fuzzy sets theory [10,11]. Second, Bayes' theorem (BT) was used as it provides a natural theoretical framework for explicitly articulating epistemic or state-of-knowledge uncertainties

in prior engineering knowledge. Such uncertainties can be updated as additional information, which becomes available from the tests and analyses conducted during a development program [12].

Kahneman and Smith [13] proposed prospect theory (PT) as a foundation for behavioral and experimental economics. The theory was updated in cumulative prospect theory (CPT) [14], where cumulative probability is converted into expected utility probability. Kahneman and Tversky [14] claimed that BT is violated when people aim to predict an uncertain outcome. For example, when investors aim to predict the future movements of a stock's price by referencing its price history, they do not consider the possibility that the history is the result of pure randomness and thus may indicate nothing meaningful. Consequently, Ali and Sanjit [15] proposed composite cumulative prospect theory (CCPT) to modify the curves of low and high probabilities in CPT. Similar to studies using BT, the research analyzed a high-probability zone. Moreover, with reference to the fuzzy utility theory (FUT), which was proposed by Kirkwood [16] to define the utility functions and in which the center-of-gravity method is adopted, these researchers used the weighted average and center of sums methods to defuzzify the influential factors and utility values [17].

In this study, a method for contractor selection is provided by integrating preference relation theory (PRT), BT, FUT, and PT. The method is executed through expert interviews as part of the recalculation approach. The present research evaluated the combination of criteria weights, probability, and utilities—as obtained through different methods—and used the Bayesian fuzzy prospect model (BFPM) for contractor selection, thus verifying whether the research's aims were met. Finally, the overall prospect values of bid commitments were calculated, which are defined in terms of the probability of the bidders implementing the commitment and the utility of such completion to the owners. The contractor with the highest score was considered the optimal applicant.

In the current method of contractor selection, committee members choose bidders in accordance with the requirements of the responsible entity. However, the selection result is not objective because the committee members have subjective preferences. Therefore, the main purpose of this study was to scientifically assess the subjective opinions of committee members. Their first impression is similar to a Bayesian prior, which is updated through external information to a Bayesian posterior probability. The score of the second impression can be obtained by BT. Moreover, the committee members have different fuzzy risk preferences for each bidder. Thus, the uncertainty of risk preferences can be presented in terms of the fuzzy utility. Subsequently, by multiplying the utility and prospect-theoretic probabilities, the expected values of each committee member are acquired as the final result of the evaluation.

## 2. Literature Review

### 2.1. Method of Selecting Multi-Criteria Decision Making for Contractor Selection

Contractor selection decisions can be made by using (1) a single-criterion decision-making model, which considers the lowest reasonable bidder, or (2) a MCDM method, in which a MCDM model is constructed by using factors related to the cost, duration, and quality [3]. A study applied MCDM to public works contractor selection in the European Union by establishing utility functions based on duration and cost factors [4].

MCDM methods have been employed in the construction industry to select project procurement systems, contractors, concessionaires, road construction, maintenance projects for investment, and dispute resolution [18]. Various new MCDM methods have been used in projects in different domains, especially management, engineering, and for different purposes, such as construction management and energy saving [19]. Fuzzy theory, BT, and utility theory have been widely used [8,9,11,12,18,20–23] to select the most appropriate contractor for building projects, with each theory involving different methods for weight calculation, such as the analytical hierarchy process (AHP), WASPAS, and TOPSIS methods [1,20,24–26]. Other MCDM methods, such as the AHP, ANP, MOORA, COPRAS, and SWARA-FUCOM, can also be applied to the problem [1,19,25,27]. The use of Grey theory,

PROMETHEE, and interval-valued intuitionistic fuzzy sets (IVIFS) in MCDM methods also aids the ranking of contractors [21,23,28–31]. MCDM methods have also been used in the selection of cultural heritage buildings [26,32–34]. The most important advantage of multi-criteria methods is their capability to weigh conflicting interests during selection [1]. The AHP is a technique that can be easily combined with other methods, and TOPSIS is a method that is algorithmically structured and easy to compute, especially when acting in combination with other techniques [7,10,22,35,36]. Table 1 summarizes existing MCDM techniques in the literature [1,7–12,18–38]. The BT-based MCDM method of recalculating the criteria weights was used in a study where the quality of a school's classes was assessed. MCDM methods such as SAW, TOPSIS, EDAS, and COPRAS were used for the evaluation [2], including the combination of Fuzzy Theory [7,10,15–17,19–21].

**Table 1.** Application of multi-criteria decision-making techniques to contractor selection in the literature.

| No | Authors | Year | Methods and Approaches |
|----|---------|------|------------------------|
| 1 | Revie and Bedford [12] | 2011 | DM and Bayes linear method (defense procuring) |
| 2 | Ferrieia, Pinheiro and Brito [34] | 2011 | Refurbishment decision support tools: A review from a Portuguese user's perspective |
| 3 | Jato-Espino, Castillo-Lopez, Rodriguez-Hernandez and Canteras-Jordana [24] | 2014 | A Review —AHP, TOPSIS . . . 22 methods (Construction) |
| 4 | Mardani, Jusoh, Nor, Khalifah, Zakwan and Valipour [1] | 2015 | A Review —AHP (32.57%), Hybrid MCDM (16.28%) Aggregation DM method (11.7%) —4.TOPIS, 5.ELECTRE, 6.ANP, 7.PROMETHEE |
| 5 | Ulubeyli and Kazaz [8] | 2016 | Fuzzy MCDM and CoSMo (subcontractor selection in international construction) |
| 6 | Stanujkic, Zavadskas, Liu, Karabasevic and Popovic [28] | 2017 | OCRA and Grey (ranking order) (investment in the most appropriate type of hotels) |
| 7 | Khanzadi, Turskis, Amiri and Chalekaee [29] | 2017 | Game theory, ADR, grey number (solve dispute resolution problems in construction) |
| 8 | Mokhtariani, Sebt and Davoudpour [33] | 2017 | Cultural heritage Building renovation —Construction marketing —Attribute Analysis: Service attributes versus construction |
| 9 | Pashaei and Moghadam [9] | 2018 | Fuzzy AHP Method (Alternative in Low-Rise Buildings) |
| 10 | Ilce and Ozkaya [25] | 2018 | AHP and MOORA methods (the raised floor choice practice consists) |
| 11 | Mardani, Jusoh, Halicka, Ejdys, Magruk and Ahmad [19] | 2018 | A review MCDM —MOORA, COPRAS, ARAS, WASPAS, SWARA —classified into 10 areas: (1) energy source, (2) buildings, (3) material, (4) project management, (5) construction management, . . . |
| 12 | Hasnain, Thaheem and Ullah [27] | 2018 | ANP-Based Decision Support System —Analytical network process (ANP) (Contractor Selection in Road Construction) |
| 13 | Liang, Zhang, Wu, Sheng and Wang, [21] | 2018 | Using Competitive and Collaborative Criteria with Uncertainty (Joint-Venture Contractor Selection) |
| 14 | Alpay and Iphar [11] | 2018 | fuzzy multi-criteria decision-making methods —Fuzzy TOPSIS and fuzzy VIKOR (Equipment selection) |

**Table 1.** *Cont.*

| | | | |
|---|---|---|---|
| 15 | Keshavarz-Ghorabaee, Amiri, Zavadskas, Turskis and Antucheviciene [22] | 2018 | A Dynamic Fuzzy Approach Based on the EDAS Method for Multi-Criteria Subcontractor Evaluation<br>—Fuzzy EDAS (MCDM Subcontractor Evaluation) |
| 16 | Ye, Zeng and Wong [37] | 2018 | Competition rule of the multi-criteria approach<br>—34 tender evaluation factors are proposed to compose the competition rule in China<br>—The composition varies slightly between public and private sectors |
| 17 | Ortiz, Pellicer and Molenaar [38] | 2018 | Management of time and cost contingencies in construction projects: a contractor perspective (a case study of two large Spanish construction companies) |
| 18 | Cao, Esangbedo, Bai and Esangbed [30] | 2019 | Contractor Selection MCDM Problem Grey<br>—SWARA-FUCOM Weighting Method (Floating Solar Panel Energy System Installation) |
| 19 | Turskis, Goranin, Nurusheva and Boranbayev [20] | 2019 | Fuzzy WASPAS and AHP methods (Determine Critical Information Infrastructures of EU Sustainable Development) |
| 20 | Antoniou and Aretoulis [18] | 2019 | TOPSIS and utility theory (highway construction contractors) |
| 21 | Morkunaite, Bausys and Zavadskas [26] | 2019 | WASPAS-SVNS Method (Contractor Selection for Sgraffito Decoration of Cultural Heritage Buildings) |
| 22 | Gunduz and Alfar [32] | 2019 | AHP Method (Innovation in project management) |
| 23 | Morkunaite, Podvezko, Zavadskas and Bausys [31] | 2019 | AHP, PROMETHEE(Ranking) (Contractor selection by Cultural heritage buildings) |
| 24 | Davoudabadi, Mousavi, Shaparauskas and Gitinavard [23] | 2019 | a new uncertain weighting and ranking based on compromise solution with linear assignment approach<br>—Interval-valued intuitionistic fuzzy sets (IVIFSs)<br>—Ranking<br>(in energy projects—A case study about the construction project selection problem) |
| 25 | Aladag and Isik [7] | 2020 | Fuzzy AHP Method (BOT project—A case study of a PPP airport project) |
| 26 | Mahamadu, Manu, Mahdjoubi, Booth, Aigbavboa and Abanda [10] | 2020 | Fuzzy TOPSIS (BIM capability assessment: Post-selection performance of organizations on construction projects) |
| 27 | Koc and Gurgun [35] | 2020 | AH P, MCDM (Contractor prequalification for green buildings—Evidence from Turkey) |
| 28 | Zhang [36] | 2020 | AHP, D-S Evidence Theory (Construction in Government public project green procurement in China) |

## 2.2. Preference Relationships Theory

Preference Relationships Theory (PRT) was adopted to calculate the relative weights between factors. The decision-making process is largely based on the preference relation for alternatives. The preference relation is a value assigned by experts to two alternatives to reflect the experts' preferences for the two alternatives. Preference relations can be applied in a decision-making model

to integrate experts' individual preferences into a group preference [39–43]. In decision making, two types of preference relations are adopted: multiplicative preference relation (MPR) and FPR [39,44].

The advantages of the combination between MPR and FPR were to develop a possibility evaluation method. First, MPR and FPR matrices were used to define linguistic variables and quantized values corresponding to linguistic variables. Subsequently, a questionnaire was administered to collect the subjective opinions of each evaluator. To integrate the experts' opinions and obtain the implementation possibility, the questionnaire results were then converted to the FPR's average weight method.

### 2.3. Bayes' Theorem

Bayes' theorem (BT) is presented in Equation (2), where the conditional probability theorem is for before or after an event [45,46].

$$p(A|B) = \frac{p(B|A) \times p(A)}{p(B)} \tag{1}$$

where $A$ and $B$ are events and $p(B) > 0$; $p(A|B)$ is the probability of event $A$ occurring if event $B$ occurs; $p(B|A)$ is the probability of event $B$ occurring if event $A$ occurs; $p(A)$: prior probability density function; $p(B)$: prior probability density function (or marginal probability function), which indicates the probability of $X$ occurring in a sample dataset; $p(B|A)$: Likelihood function, sample distribution; $p(A|B)$: Posterior probability density function.

1.  The prior probability is expressed as a cumulative distribution function (CDF) as follows:

$$w_1(p) = exp\left(-\beta(-lnp)^{\alpha}\right) \tag{2}$$

   where $w_1(p)$ is prior probability density function; $\alpha$ and $\beta$ are the parameters of Equation (2).
2.  The maximum likelihood distribution depends on additional information.
3.  The posterior probability ($w_2(p)$) is given as follows:

$$w_2(p) = exp\left(-\beta(-lnp)^{\alpha}\right)$$

4.  Posterior probability = prior probability × likelihood function. The posterior probability is expressed as a probability density function (PDF) as follows:

$$w_2(p) = \frac{w_1(p) \times L(p)}{\sum w_1(p) \times L(p)} \tag{3}$$

### 2.4. Prospect Theory

Expected utility theory postulates that all possible outcomes of an uncertain event have their respective utilities and probabilities, where the sum of the utility–probability product of each possible outcome represents the expected utility of an uncertain event. Subsequently, PT adopts probability weighting functions to explain the non-linear preferences of people when they evaluate the probability of uncertain events [47] (pp. 282–283). Specifically, every result corresponds to the product of decision weights (subjective probability) and psychological values (utility), and this product represents the decision prospect value.

1.  PT and investment psychology (a four-level model): Kahneman and Smith [13] proposed the S-shaped utility function in Foundations of Behavioral and Experimental Economics. It has experimentally indicated that people have non-linear preferences when evaluating probabilities [35]. This preference is characterized by (1) a tendency for "loss aversion," in which a unit loss is perceived to be of a greater magnitude than a unit gain; (2) a tendency for "risk aversion" in gain situations; (3) a tendency for "risk seeking" in loss situations; and (4) a tendency to make decisions based on a "reference point" to determine gain or loss situation.

2.  CPT: This theory uses a cumulative probability to convert an expected utility probability. Tversky and Kahneman [14,48] and Prelec [49] have proposed different parameters for the probability weighting function.
3.  CCPT: Ali and Dhami [15] (pp. 14–16) proposed a probability weighting function which uses the composite Prelec probability weighting function (CPF) [49] to correct the curve function of high- and low-probability zones. Due to this correction, changes in subjective decision-making probabilities after the provision of external information can be better reflected.

## 2.5. Influence Factors Considered

In the European 2020 strategy, the EU mentioned three trends that strengthen economic and social development: "smart growth", "sustainable growth", and "inclusive growth". For the sustainable growth of the construction industry, Taiwan must meet the "sustainable public engineering indicators" issued by the Public Works Commission: safety, creativity, humanities, durability, waste reduction, energy saving, ecology and benefits.

A review of the literature on the following topics was conducted: contractor selection criteria [50,51], the key determinants of contractor performance, and the relative importance of quality assessment [10,20,26,27,35,36,52–60]; 18 references and 30 influencing factors were compiled (See Table A1). In addition, 22 influencing factors belong to sustainable development criterions, indexes in critical information infrastructure, cultural heritage, or green procurement [20,26,36].

## 2.6. Summary

According to the literature review above, the merits of PRT, BT, FUT, and PT were combined to develop BFPM for the use in a contractor selection MCDM. Utility theory and fuzzy statistical methods in PRT were used to model the uncertainties of qualitative factors and the risk preferences of subjective utilities. The use of PRT addressed consistency-related issues in the pairwise comparison matrix of the AHP method. PRT, BT, and FUT were integrated to represent expert knowledge on state evaluations and to model the expected values that can be acquired using the product of probability and utility. BT, an effective method for assessing the posterior probabilities of the influential factors once additional information was provided, was applied to simulate the probability of success, which was then used to determine the probability that bid commitment was implemented. The strengths of BFPM are to help owner to make the optimal contractor selection decision with taking not only the utilities of bid contractors but also the probability of bid implementation into account. Therefore, BFPM was used to model the transformation of risk or uncertainty in contractor selection, capturing the difference relationship between MCDM and BFPM. A public project includes a concept of life cycle risk management, and only after facilities are smoothly and successfully completed, they can be operated and maintained. The entire project life cycle incorporates four main stages, including planning and design, project bidding, project performance, and operation and maintenance, as well as each stage carefully selecting contractors to handle the corresponding tasks. When each stage has selected its contractor, through the series of works in each stage, government can ensure the quality of the public buildings or facilities, reduce maintenance costs, increase facility efficiency, and control energy consumption.

## 3. Constructing the Bayesian Fuzzy Prospect Model

This paper presents a decision-making procedure for selecting a general contractor for construction projects. Moreover, this study examined the duration discount, cost discount, and quality assurance, which are the three influence factors belonging to the sustainable development indicators in a contractor's bid commitment. The duration discount, $d$, is defined as the proportion of the difference between committee members' expected duration and the bid duration to committee members' budgeted duration. The cost duration, $c$, is defined as the proportion of the difference between committee members' expected cost and the bid cost to the owner's budgeted cost. The quality assurance, $q$,

is defined as the proportion of the difference between the bid commitment and committee members' or owners' quality requirement. Each commitment is assumed to have two possible outcomes for the contract performance after a bid is won: success or failure (in the implementation). Probability theory assumes that these two possible outcomes can occur. The terms $(pd_S, pd_F)$, $(pc_S, pc_F)$, and $(pq_S, pq_F)$ represent the possibilities of the two outcomes of bid commitment in terms of the duration discount, cost discount, and quality assurance, respectively, and are collectively referred to as "the implementation probability of bid commitment."

The case example of the present study is a construction project involving mass rapid transit station development. Through the Bayesian probability evaluation method, the implementation possibility for the case study was obtained. The duration of the project was 40 months, and the budgeted cost was 199 million Taiwan Dollars (TWD).

The FPR matrix was adopted to construct an evaluation model. This model enabled committee members to identify the relative importance of factors and to evaluate the implementation possibility of bid commitment. Subsequently, the evaluation model was integrated with PT to develop a decision-making model. The Bayesian probability model and fuzzy PT comprised four parts. The stage for constructing the Bayesian fuzzy prospect model (BFPM) is shown in Figure 1.

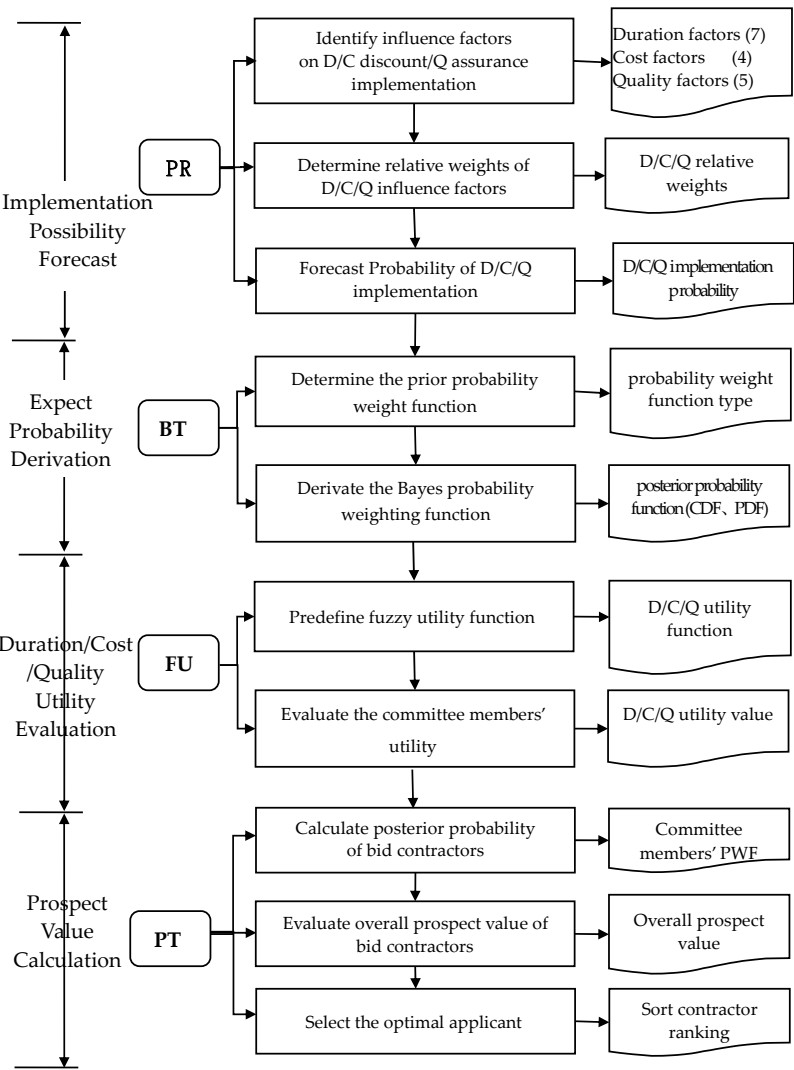

**Figure 1.** Constructing the Bayesian fuzzy prospect model (BFPM).

### 3.1. Asssessment for Implementation Possibility

Preference relationship theory was used to (1) evaluate the influential factors for selecting a contractor, (2) determine the relative weights between influential factors, and (3) evaluate the implementation possibility of bid commitment. First, the duration, cost, and quality factors were determined through a literature review and organized into factors recommended for use in this study. Subsequently, the MPR and FPR were used to calculate the weights of the duration, cost, and quality factors. The MPR and FPR were also employed to evaluate the implementation possibility of bid commitment according to two possible outcomes: success and failure (in the implementation).

### 3.2. Derivation of Expected Probability

BT was used to determine (1) the prior probability weighting function and (2) the Bayesian probability weighting function. Due to external environmental information, the prior probability weighting function was obtained by using the CPT probability function and the parameter value. The posterior probability, which was based on the CCPT, was used to derive a likelihood function that satisfied Bayes' theorem. This function was used as the expected probability of a contractor being selected by committee members.

### 3.3. Evaluation of Utilities for Duration, Cost and Quality

FU theory was used to (1) determine the fuzzy utility function (FUF) and (2) evaluate the utility of bid commitment for the committee members. To determine the FUF, we adopted the FUF proposed by Kirkwood [16] and referred to the utility function provided by Cheng and Kang [17]. Expert questionnaires were collected and organized, and the FUF was established. Subsequently, the center-of-gravity method was employed to evaluate the differences in utility between potential contractors. After the duration discount, cost discount, and quality assurance (%) were converted, the FUF was used to calculate the utility of bid commitment for the committee members.

### 3.4. Overall Prospect Evaluation of Candidate Contractors

CCPT was used to (1) calculate the posterior probability of bid contractors, (2) evaluate the overall prospect value of bid contractors, and (3) select the optimal contractor. The posterior probability of bid contractors was calculated by evaluating the relevant expected probability of committee members and then multiplying it with the utility of bid commitment to obtain the overall prospect value of a potential contractor for reference. After the contractors were ranked, the optimal and runner-up applicants were selected. Finally, the contractor selection results obtained from BFPM were compared with the lowest tender and MCDM (overall utility values [4]), as well as the multi-criteria prospect model (MCPM) results.

## 4. Assessment Implementation Possibility for Bid Commitment

### 4.1. Identifying Influence Factors of Duration, Cost and Quality Implementation

Sixteen influence factors were selected, of which, seven ($DF_k$, $k = 1, 2, \ldots, 7$) were used for the duration discount, four ($CF_k$, $k = 1, 2, 3, 4$) were used in cost discount implementation, and the remaining five ($QF_k$, $k = 1, 2, 3, 4, 5$) were used in quality assurance implementation. Among them, 5 influence factors belong to sustainable growth indexes, including $DF1$, $DF4$, $QF1$, $QF2$ and $QF3$. Technical ability (DF1) refers to creativity; plan management ($DF4$) refers to safety, health, and environment protection; building materials (capacity)/equipment resources ($QF1$) refers to green building mark; after sales service ($QF2$) refers to the feedback facility about humanities; and warranty period ($QF3$) refers to waste reduction and energy saving. $QF2$ and $QF3$ of influence factors can achieve sustainability in the operation stage of construction life cycle (see Table 2 for details).

**Table 2.** Influence factors of the duration, cost, and quality in bid commitment.

| Item | Influence Factors | No. |
|---|---|---|
| **Duration** | • Technical ability(creativity) <br> • Manufacturer qualification manpower <br> • Planning and Control <br> • Plan Management (Safety and health, environment protection) <br> • Financial status (capacity) <br> • Construction period or delivery capacity <br> • Labor relations (resolving conflicts) | DF1 <br> DF2 <br> DF3 <br> DF4 <br> DF5 <br> DF6 <br> DF7 |
| **Cost** | • Contract execution volume <br> • Goodwill and the industry's greatest position <br> • Historical performance <br> • Price (cost) | CF1 <br> CF2 <br> CF3 <br> CF4 |
| **Quality** | • Building materials (capacity)/equipment resources (Green building mark) <br> • After-sales service (Feedback facility about humanities) <br> • Warranty period (waste reduction, energy saving) <br> • Management organization (control) <br> • Communication cooperation / subcontracting situation | QF1 <br> QF2 <br> QF3 <br> QF4 <br> QF5 |

### 4.2. Determining Relative Weights between Influence Factors

The MPR of an alternative set X can be expressed using matrix A = ($a_{ij}$), where ($a_{ij}$) represents the intensity of a preference for alternative $x_i$ relative to alternative $x_j$. In other words, the intensity of a preference for alternative $x_i$ is $a_{ij}$ times that for alternative $x_j$. In addition, the diagonal matrix of the MPR matrix has a multiplicative reciprocal relationship. According to Satty, the multiplicative reciprocal MPR matrix A = ($a_{ij}$) must be consistent, that is, $a_{ij} \cdot a_{jk} = a_{ik}, \ldots, \forall i, j, k \in \{1, \ldots, n\}$.

Table 3 presents the linguistic variables defined according to fuzzy theory and MPR. These variables can be used by evaluators or decision makers to indicate the relative importance of each factor and their preference for the possibility of duration/cost/quality (D/C/Q) commitment being implemented.

**Table 3.** Definition of linguistic variables.

| Linguist Variables | Very Low | Low to Very Low | Low | Fair to Low | Fair | Fair to High | High | High to Very High | Very High |
|---|---|---|---|---|---|---|---|---|---|
| Symbol | VL | LVL | L | FL | F | FH | H | HVH | VH |
| Quantitative value | 1/5 | 1/4 | 1/3 | 1/2 | 1 | 2 | 3 | 4 | 5 |

This study defined simple linguistic terms quantified on the scale [1/5, 1/4, 1/3, 1/2, 1, 2, 3, 4, 5], with each item symbolized by VL, LAL, L, FL, F, FH, H, HVH, and VH, respectively. This scale allows evaluators to express the relative degree of importance and implementation probability of the D/C/Q commitment.

For example, if the importance of $DF_1$ relative to $DF_2$ is "very high," then $DF_1$ is five times more important than $DF_2$ is.

By using the same set of linguistic variables, the possibility of "success implementation" for "failure implementation" (represented by S/F) and the possibility of "failure implementation" for

"success implementation" (represented by F/S) are evaluated according to the bidder's duration, cost, and quality commitments. Details are presented in Section 4.3.

$x$ times on the seven factors that influence the implementation of the duration discount $DF_k$, $k = 1, 2, \ldots, 7$. A questionnaire, answered by six experts with experience in contractor selection, was then used to evaluate the relative importance of each factor. We have drawn up the statement, and the experts agreed to adopt anonymity and make the questionnaire contents public in the beginning of questionnaire. The questionnaire results of Evaluator No.1 are presented in Figure 2 as an example.

|  | DF$_1$ | DF$_2$ | DF$_3$ | DF$_4$ | DF$_5$ | DF$_6$ | DF$_7$ |
|---|---|---|---|---|---|---|---|
| DF$_1$ | 1 | H |  |  |  |  |  |
| DF$_2$ | $a(df)_{12}^{-1}$ | 1 | FH |  |  |  |  |
| DF$_3$ |  | $a(df)_{23}^{-1}$ | 1 | F |  |  |  |
| DF$_4$ |  |  | $a(df)_{34}^{-1}$ | 1 | F |  |  |
| DF$_5$ |  |  |  | $a(df)_{45}^{-1}$ | 1 | FH |  |
| DF$_6$ |  |  |  |  | $a(df)_{56}^{-1}$ | 1 | H |
| DF$_7$ |  |  |  |  |  | $a(df)_{67}^{-1}$ |  |

**Figure 2.** Multiplicative preference pairwise comparison matrix (for Evaluator 1).

In Figure 2, H indicates that DF$_1$ is more important than DF$_2$ (specifically three times more important than DF$_2$, as stated in Table 3). The evaluator's preference corresponding to the six selected symbols based on the definition of linguistic variables (Table 3) was {3, 2, 1, 1, 2, 3}. These six preference values are expressed using the corresponding set of variables {$a(df)_{12}$, $a(df)_{23}$, $a(df)_{34}$, $a(df)_{45}$, $a(df)_{56}$, $a(df)_{67}$}.

The MPR matrix was constructed through four steps. The matrix was used to analyze the relative importance between the factors that influence the implementation of the duration discount {$DF_1, \ldots, DF_7$}. The steps in the matrix construction were as follows:

Step 1. Use Equation (4) to calculate all preference values for set $B$ (the set of preference values).

$$B = \left\{ a(df)_{ij}, i < j \wedge a(df)_{ij} \notin \{a(df)_{12}, a(df)_{23, \ldots, a(df)_{67}}\} \right\},$$
$$a(df)_{ij} = a(df)_{ii+1} \times a(df)_{i+1i+2}, \ldots, \times a(df)_{j-1j}. \tag{4}$$

For example, $a(df)_{17} = (df)_{12} \times a(df)_{23} \times a(df)_{34} \times a(df)_{45} \times a(df)_{56} \times a(df)_{67} = 3 \times 2 \times 1 \times 1 \times 2 \times 3 = 36$, $a(df)_{27} = a(df)_{23} \times a(df)_{34} \times a(df)_{45} \times a(df)_{56} \times a(df)_{67} = 2 \times 1 \times 1 \times 2 \times 3 = 12$ [17].

Step 2: Construct the following MPR matrix.

$$A = \left\{ a(df)_{12}, a(df)_{23, \ldots, a(df)_{67}} \right\} \cup B \cup \left\{ a(df)_{12}, a(df)_{23, \ldots, a(df)_{67}} \right\}^{-1} \cup B^{-1} \tag{5}$$

where $A$ is the MPR for the relative weights of $dfk$; $B^{-1} = \{1/a(tf)_{ji}\}$. For example, $a(df)_{71} = 1/36$, $a(df)_{72} = 1/12$.

Step 3: Identify the maximum value in Matrix $A$:

$$z = \max A \tag{6}$$

Step 4: Convert Matrix A into a consistent MPR matrix $C = f(A)$, as presented in Equation (7).

$$f: [1/z, z] \rightarrow [1/5, 5], c(df)_{ij} = a(df)_{ij}^{1/\log_5 z}, \text{ for } z > 5; c_{ij} = a_{ij}, \text{ for } z < 5 \tag{7}$$

For example, $z = 36$, $1/\log_5 36 = 0.44912$, $c(df)_{17} = 36^{0.44912} = 5$, $c(df)_{71} = 1/5$.

Equations (4)–(7) were used to construct and complete the MPR matrix that reflects the relative importance between the factors that influence the implementation of the duration discount {$DF_1, \ldots, DF_7$}.

The FPR of alternative set *X* can be expressed using matrix $P = [pij]$, where $[pij]$ represents the intensity of preferences for alternative *xi* relative to alternative *xj*, and *pi* is between 0 and 1, as determined by the fuzzy membership function. A *pij* value of 0.5 indicates indifference between *xi* and *xj*; *pij* = 1 indicates that *xi* is highly preferred over *xj*; *pij* = 0 indicates that *xj* is highly preferred over *xi*. In addition, the diagonal matrix of the FPR matrix is assumed to be an additive reciprocal matrix, that is, $pij + pji = 1, \forall i, j \in \{1, \ldots, n\}$.

According to the definition of the consistent FPR matrix [61], the domain range was [1/5, 5]. Moreover, the consistent MPR matrix C can be converted into an FPR matrix with a domain range of [0, 1] [5] by using the function $D = [d(df)_{ij}]$, where

$$d(df)_{ij} = 0.5 \times (1 + \log_5 c(df)_{ij}) \tag{8}$$

For example, $d(df)_{17} = 0.5 \times (1 + \log_5 5) = 1$, $d(df)_{71} = 0.5 \times (1 + \log_5 1/5) = 0$.

By using Equation (8), the six evaluators' FPR matrices $(D_1, D_2, \ldots, D_6)$ were converted, and the average FPR matrix E is given as follows:

$$E = (D_1 + D_2 + \ldots + D_6)/6 \tag{9}$$

The average FPR matrix $E = [e(df)_{ij}]$ was calculated to obtain the normalized FPR matrix $Q = [q(df)_{ij}]$, which is expressed as follows.

$$q(df)_{ij} = e(df)_{ij} \Big/ \sum_{i=1}^{7} e(df)_{ij} \tag{10}$$

The FPR matrix is presented in Table 4, where the relative weights $r(df)i$ for the seven influence factors of duration discount were obtained through the following Equation (11) and the relative weights $r(df)i$ for the four influence factors of duration discount were calculated as 0.18, 0.13, 0.11, 0.15, 0.16, 0.16 and 0.11 in Table 4.

$$r(df)_i = \sum_{j=1}^{7} q(df)_{ij} \Big/ \sum_{i=1}^{7} \sum_{j=1}^{7} q(df)_{ij} \tag{11}$$

**Table 4.** Relative weight fuzzy preference relation matrix of the influence factors of duration discount.

| $q(df)_{ij}$ | DF$_1$ | DF$_2$ | DF$_3$ | DF$_4$ | DF$_5$ | DF$_6$ | DF$_7$ | Sum $\Sigma q(df)_{ij}$ | Relative Weights $r(df)_i$ |
|---|---|---|---|---|---|---|---|---|---|
| DF$_1$ | 0.20 | 0.18 | 0.17 | 0.18 | 0.19 | 0.19 | 0.17 | 1.29 | 0.18 |
| DF$_2$ | 0.12 | 0.13 | 0.13 | 0.13 | 0.12 | 0.13 | 0.13 | 0.89 | 0.13 |
| DF$_3$ | 0.10 | 0.11 | 0.12 | 0.11 | 0.10 | 0.11 | 0.12 | 0.77 | 0.11 |
| DF$_4$ | 0.15 | 0.15 | 0.15 | 0.15 | 0.15 | 0.15 | 0.15 | 1.03 | 0.15 |
| DF$_5$ | 0.18 | 0.16 | 0.16 | 0.17 | 0.17 | 0.17 | 0.16 | 1.18 | 0.16 |
| DF$_6$ | 0.16 | 0.16 | 0.16 | 0.16 | 0.16 | 0.16 | 0.15 | 1.11 | 0.16 |
| DF$_7$ | 0.09 | 0.11 | 0.11 | 0.11 | 0.10 | 0.10 | 0.11 | 0.74 | 0.11 |
| | | | | | | | Overall | 7.00 | 1.00 |

By using the aforementioned steps, the relative weights $r(cf)_i$ for the four influence factors of cost discount were calculated as 0.30, 0.24, 0.23, and 0.23. Subsequently, the relative weights $r(qf)_i$ for the five influence factors of quality assurance were calculated as 0.27, 0.17, 0.14, 0.20, and 0.22.

### 4.3. Assessing the Probability of Fulfilling the Bid Commitment

With reference to the prediction results of Wang and Chang [62] (pp. 807–810) on the possibility of successful knowledge management implementation, we formulated the following steps to evaluate the implementation possibility of the D/C/Q commitments of a bid.

Step 1: Define linguistic variables and design the questionnaire.

Every candidate contractor was required to provide evidence relating to factors affecting $DF_k$, $CF_k$, and $QF_k$ implementation. Then, with respect to $DF_k$, evaluators El to E6 were asked to select one preference linguistic term for success over failure.

Step 2: The evaluators completed the questionnaire by referring to the evidence on each factor.

The implementation possibility of contractor A's duration commitment was evaluated. Table 5 presents the six evaluators' questionnaire results.

**Table 5.** Questionnaire results for evaluating implementation possibility (for contractor A).

| Duration Factor | S | Linguist Variables | | | | | | | | | F | | | E1 | E2 | E3 | E4 | E5 | E6 |
|---|---|---|---|---|---|---|---|---|---|---|---|---|---|---|---|---|---|---|---|
| | | VH | HVH | H | FH | F | FL | L | LVL | VL | | | | F | F | F | F | F | F |
| DF$_1$ | | $v$ | | | | | | | | | | DF$_1$ | S | 5 | 4 | 4 | 3 | 3 | 4 |
| DF$_2$ | | | $v$ | | | | | | | | | DF$_2$ | S | 4 | 5 | 4 | 4 | 3 | 3 |
| DF$_3$ | | | | $v$ | | | | | | | | DF$_3$ | S | 3 | 3 | 3 | 3 | 3 | 3 |
| DF$_4$ | | $v$ | | | | | | | | | | DF$_4$ | S | 5 | 4 | 5 | 4 | 5 | 3 |
| DF$_5$ | | | | $v$ | | | | | | | | DF$_5$ | S | 3 | 5 | 3 | 3 | 5 | 3 |
| DF$_6$ | | | $v$ | | | | | | | | | DF$_6$ | S | 4 | 3 | 5 | 4 | 4 | 3 |
| DF7 | | | | $v$ | | | | | | | | DF$_7$ | S | 3 | 5 | 4 | 5 | 4 | 4 |

(Left block labelled vertically "Success Implementation"; right block labelled vertically "Failure Implementation".)

$DF_1, \ldots, DF_7$: the seven influence factors for duration discount; $E_1, \ldots, E_6$: the evaluators; **S/F**: evaluated probability of assessing "successful implementation" over "failed implementation" for a given influence factor.

For each influence factor, the evaluators performed a relative comparison of S/F and obtained quantized values according to the definition of linguistic variables.

Step 3: Convert quantized values into FPR values.

The MPR value $ads_{kl} \in \left[\frac{1}{5}, 5\right]$ was converted into a consistent FPR value $bds_{kl} \in [0, 1]$.

The terms $bds_{kl} = \frac{1}{2}(1 + log_5 ads_{kl})$ and $cds_k = \frac{1}{m}\sum_{l=1}^{m} bds_{kl}$ were calculated, where cds$_k$ is the preference rating success/failure related to DF$_k$.

Step 4: Determine the group fuzzy preference value.

The mean quantized value of the six evaluators was expressed as $cds_k$, where $k = 1, \ldots, 7$. This value can be used to evaluate the implementation possibility of duration commitment (Table 6). The analogous mean values $ccs_k$, where $k = 1, \ldots, 4$, and $cqs_k$, where $k = 1, \ldots, 5$, can be used to evaluate the implementation possibilities of cost commitment and quality commitment, respectively.

**Table 6.** Convergence fuzzy preference value for the assessment group.

| | | E1 | E2 | E3 | E4 | E5 | E6 | Average |
|---|---|---|---|---|---|---|---|---|
| | | F | F | F | F | F | F | Cdsk |
| DF$_1$ | S | 1.00 | 0.93 | 0.93 | 0.84 | 0.84 | 0.93 | 0.91 |
| DF$_2$ | S | 0.93 | 1.00 | 0.93 | 0.93 | 0.84 | 0.84 | 0.91 |
| DF$_3$ | S | 0.84 | 0.84 | 0.84 | 0.84 | 0.84 | 0.84 | 0.84 |
| DF$_4$ | S | 1.00 | 0.93 | 1.00 | 0.93 | 1.00 | 0.84 | 0.95 |
| DF$_5$ | S | 0.84 | 1.00 | 0.84 | 0.84 | 1.00 | 0.84 | 0.89 |
| DF$_6$ | S | 0.93 | 0.84 | 1.00 | 0.93 | 0.93 | 0.84 | 0.91 |
| DF$_7$ | S | 0.84 | 1.00 | 0.93 | 1.00 | 0.93 | 0.93 | 0.94 |

Step 5: Normalize a series of FPRs to calculate the assessment values corresponding to each factor.

Each consistent FPR matrix was normalized using the Equation (12) to derive the normalized FPR matrix $Q_k$.

$$Q_K = \left[qt_{ij}\right]_k ; \; qt_{ij} = dt_{ij}\bigg/\sum_{i=1}^{2} dt_{ij} \tag{12}$$

Step 6: Combine the weights of the influence factor to determine the implementation possibility.
The normalized FPR matrix and Equation (13) can be used in the analysis to obtain a set of two degrees of implementation $[rd_1, rd_2]$ (Table 7).

$$rd_i = \sum_{j=1}^{2} qd_{ij} \Bigg/ \sum_{i=1}^{2} \sum_{j=1}^{2} qd_{ij} \tag{13}$$

**Table 7.** Constructing the multiplicative preference relation matrix for each factor.

|        |         | Success | Failure |        |   | S    | F    | Sum  | Evaluating Value |
|--------|---------|---------|---------|--------|---|------|------|------|------------------|
| $DF_1$ | Success | 0.50    | 0.91    | $DF_1$ | S | 0.91 | 0.65 | 1.55 | 0.78             |
|        | Failure | 0.05    | 0.50    |        | F | 0.09 | 0.35 | 0.45 | 0.22             |
|        | Overall | 0.55    | 1.41    |        | Overall |  |  | 2.0  |                  |
| $DF_2$ | Success | 0.50    | 0.91    | $DF_2$ | S | 0.91 | 0.65 | 1.55 | 0.78             |
|        | Failure | 0.05    | 0.50    |        | F | 0.09 | 0.35 | 0.45 | 0.22             |
|        | Overall | 0.55    | 1.41    |        | Overall |  |  | 2.00 |                  |
| $DF_3$ | Success | 0.50    | 0.84    | $DF_3$ | S | 0.76 | 0.63 | 1.38 | 0.69             |
|        | Failure | 0.16    | 0.50    |        | F | 0.24 | 0.37 | 0.62 | 0.31             |
|        | Overall | 0.66    | 1.34    |        | Overall |  |  | 2.00 |                  |
| $DF_4$ | Success | 0.50    | 0.95    | $DF_4$ | S | 0.96 | 0.66 | 1.62 | 0.81             |
|        | Failure | 0.02    | 0.50    |        | F | 0.04 | 0.34 | 0.38 | 0.19             |
|        | Overall | 0.52    | 1.45    |        | Overall |  |  | 2.00 | 1.00             |
| $DF_5$ | Success | 0.50    | 0.89    | $DF_5$ | S | 0.82 | 0.64 | 1.46 | 0.73             |
|        | Failure | 0.11    | 0.50    |        | F | 0.18 | 0.36 | 0.54 | 0.27             |
|        | Overall | 0.61    | 1.39    |        | Overall |  |  | 2.00 | 1.00             |
| $DF_6$ | Success | 0.50    | 0.91    | $DF_6$ | S | 0.86 | 0.65 | 1.51 | 0.75             |
|        | Failure | 0.08    | 0.50    |        | F | 0.14 | 0.35 | 0.49 | 0.25             |
|        | Overall | 0.58    | 1.41    |        | Overall |  |  | 2.00 | 1.00             |
| $DF_7$ | Success | 0.50    | 0.94    | $DF_7$ | S | 0.79 | 0.65 | 1.45 | 0.72             |
|        | Failure | 0.13    | 0.50    |        | F | 0.21 | 0.35 | 0.55 | 0.28             |
|        | Overall | 0.63    | 1.44    |        | Overall |  |  | 2.00 | 1.00             |

Corresponding to the seven influence factors of duration discount $DF_k$, where $k = 1, 2, \dots, 7$, the seven sets of implementation degrees were grouped to form a possibility analysis matrix $[r(df)_k]$. Subsequently, the relative importance matrix $[r(df)_k]$ of the seven factors $DF_{k,}$, where $k = 1, 2, \dots, 7$, which was obtained using Equation (8), was calculated to determine the possibilities of the two outcomes of duration discount as follows:

$$pd_i = \sum_{k=1}^{7} \left( rd_{ik} \times r(df)_k \right) \tag{14}$$

where *pdi* is the probability on implementation of *i*'s time.

The successful implementation probability of contractor A's duration discount was evaluated. Table 8 presents the analysis results. To evaluate the successful implementation probabilities for cost discount and quality assurance, the steps mentioned above were repeated. The results are presented in Table 7. In addition, the relative weights between each influence factor obtained in Section 4.2 were multiplied. The results indicated that contractor A had success implementation probabilities of 0.76, 0.68, and 0.72 for D/C/Q commitments, respectively. The aforementioned method can be used to determine the successful implementation probabilities of the D/C/Q commitments of other contractors.

**Table 8.** Successful implementation probability of a bid's duration/cost/quality (D/C/Q) commitments (for contractor A).

| Duration Commitment Section | | | Cost Commitment Section | | | Quality Commitment Section | | |
|---|---|---|---|---|---|---|---|---|
| Influence Factor | Relative Weights | Success Implementation (s) | Influence Factor | Relative Weights | Success Implementation (s) | Influence Factor | Relative Weights | Success Implementation (s) |
| $DF_1$ | 0.18 | 0.78 | CF1 | 0.30 | 0.70 | QF1 | 0.27 | 0.72 |
| $DF_2$ | 0.13 | 0.78 | CF2 | 0.24 | 0.67 | QF2 | 0.17 | 0.74 |
| $DF_3$ | 0.11 | 0.69 | CF3 | 0.23 | 0.63 | QF3 | 0.14 | 0.74 |
| $DF_4$ | 0.15 | 0.81 | CF4 | 0.23 | 0.73 | QF4 | 0.20 | 0.74 |
| $DF_5$ | 0.16 | 0.73 | | | | QF5 | 0.22 | 0.69 |
| $DF_6$ | 0.16 | 0.76 | | | | | | |
| $DF_7$ | 0.11 | 0.72 | | | | | | |
| Success robability ($pds$) | | 0.76 | Success probability ($pcs$) | | 0.68 | Success probability ($pqs$) | | 0.72 |

## 5. Assessment of the Committee Members' Expected Probability

By using BT, the assessment group first determined the prior and posterior cumulative probability weight function. Among these functions, the prior probability weight function is the CPT probability function and parameter value. Because of external environmental information, the posterior probability was based on CCPT. BT was used to convert the CDF into a PDF to derive the likelihood function $L(p)$. This function was a normal distribution, which indicated that the CCPT is in agreement with BT. In addition, the result can be regarded as the expected probability that committee members will select a contractor. The derivation process of the Bayesian probability weighting function is illustrated in Figure 3.

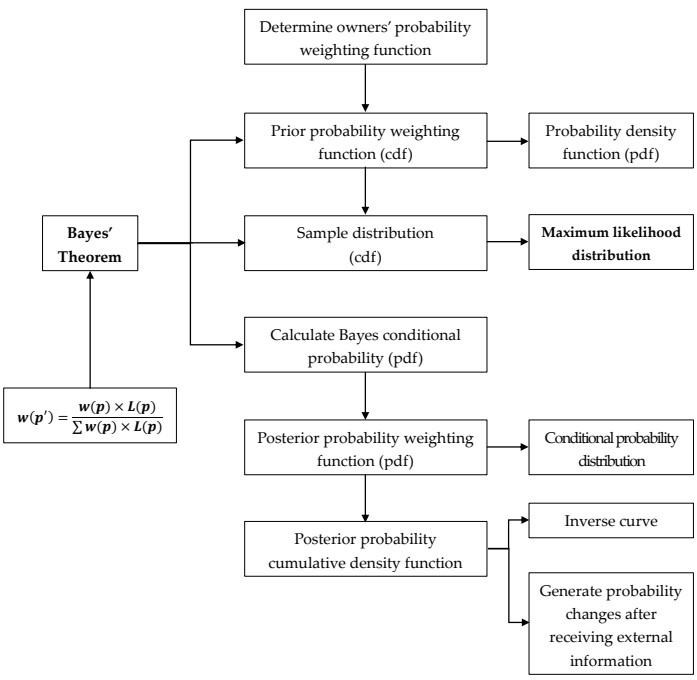

**Figure 3.** Derivation of the Bayesian probability weighting function.

### 5.1. Determination of the Prior Probability Weight Function

This research used CPT probability weighting function [45] as the prior probability weight function (see Figure 4). Subsequently, according to the parameter results calculated by Cheng and Kang through

a questionnaire, namely $\alpha = 0.62$ and $\beta = 0.97$ [17] (p. 1059), the prior probability function was obtained using the following equation:

$$w(p) = exp(-\beta(-lnp)^{\alpha}) = exp\left(-0.97(-lnp)^{0.62}\right)$$

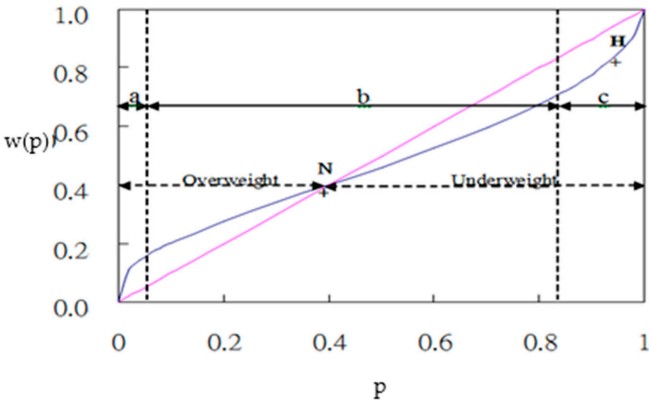

**Figure 4.** Prior probability function. (Source: Tversky and Kahneman [48]).

### 5.2. Derivation of the Bayesian Probability Weight Function

CCPT modifies the curves of low probability and high probability in CPT [15]. The Prelec function, which is presented in the middle part of Figure 5, is usually not conformed to in BT and other relevant theories of probability in which uncertain results are expected [9]. As the decision maker considers tenders with high probability as priority contractors, the present research focused on the high-probability zone ($p = 0.66–1$). The posterior probability of BT was set as the risk probability (Figure 5).

$$\underline{p} = e^{-\left(\frac{\beta}{\beta_0}\right)^{\frac{1}{\alpha_0-\alpha}}}, \overline{p} = e^{-\left(\frac{\beta}{\beta_1}\right)^{\frac{1}{\alpha_1-1}}} \tag{15a}$$

$$\omega(p) = \begin{cases} 0 & if \quad p = 0 \\ e^{-\beta_0(-lnp)^{\alpha_0}} & if \quad 0 < p \le \underline{p} \\ e^{-\beta(-lnp)^{\alpha}} & if \quad \underline{p} < p \le \overline{p} \\ e^{-\beta_1(-lnp)^{\alpha_1}} & if \quad \overline{p} < p \le 1 \end{cases} \tag{15b}$$

$$0 < \alpha < 1, \beta > 0; \alpha_0 > 1, \beta_0 > 0; \alpha_1 > 1, \beta_1 > 0, \beta_0 < 1/\beta^{\frac{\alpha_0-1}{1-\alpha}}, \beta_1 > 1/\beta^{\frac{\alpha_1-1}{1-\alpha}} \tag{15c}$$

$$p_1 = e^{-\left(\frac{1}{\beta_0}\right)^{\frac{1}{\alpha_0-1}}}, \quad p_2 = e^{-\left(\frac{1}{\beta}\right)^{\frac{1}{\alpha-1}}}, \quad p_3 = e^{-\left(\frac{1}{\beta_1}\right)^{\frac{1}{\alpha_1-1}}} \tag{15d}$$

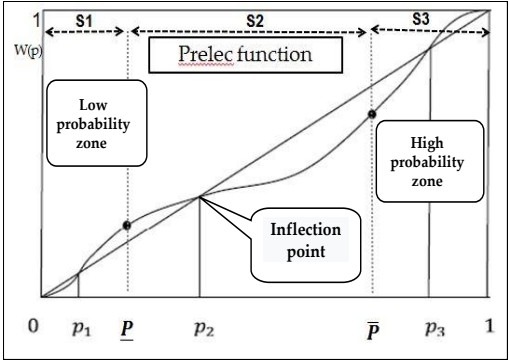

**Figure 5.** Posterior probability function. (Source: Ali and Dhami [15]).

Equation (15b) $e^{-\beta(-lnp)^{\alpha}}$ was written in Section 5.1; therefore, $e^{-\beta(-lnp)^{\alpha}} = e^{-0.97(-lnp)^{0.62}}$, such that $\alpha = 0.62$, $\beta = 0.97$.

1.  Solve $p_1$.
    Let $p_1 = e^{-\beta_0(-lnp)^{\alpha_0}}$.
    Take ln on both sides; $p_1 = -\beta_0(-lnp)^{\alpha_0}$.
    Take exp on both sides; $p_1 = e^{-(\frac{1}{\beta_0})^{\frac{1}{\alpha_0-1}}}$.

2.  Similarly, obtain $\underline{p} = e^{-(\frac{\beta}{\beta_0})^{\frac{1}{\alpha_0-\alpha}}}$ and $p_3 = e^{-(\frac{1}{\beta_1})^{\frac{1}{\alpha_1-1}}}$.

3.  Let the prior probability function [17] (pp. 45–48) be $e^{-\beta(-lnp)^{\alpha}} = e^{-0.97(-lnp)^{0.62}}$
    Let $p_1 = 0.1$, $p_2 = 0.4$, $p_3 = 0.83$, $\underline{p} = 0.2$, and $\overline{p} = 0.66$.

    Using the equations $p_1 = e^{-(\frac{1}{\beta_0})^{\frac{1}{\alpha_0-1}}}$ and $\underline{p} = e^{-(\frac{\beta}{\beta_0})^{\frac{1}{\alpha_0-\alpha}}}$, calculations were conducted.

    $$e^{-\beta_0(-lnp)^{\alpha_0}} = e^{-0.61(-lnp)^{1.59}}$$

4.  Using the equations $p_3 = e^{-(\frac{1}{\beta_1})^{\frac{1}{\alpha_1-1}}}$ and $\overline{p} = e^{-(\frac{\beta}{\beta_1})^{\frac{1}{\alpha_1-1}}}$ in the calculation, the following result was obtained:

    $$e^{-\beta_1(-lnp)^{\alpha_1}} = e^{-1.89(-lnp)^{1.38}}$$

The aforementioned parameter estimation results met the requirement of Equation (15b). In addition, the posterior probability function of the high-probability zone can be obtained as follows:

$$e^{-\beta_1(-lnp)^{\alpha_1}} = e^{-1.89(-lnp)^{1.38}}$$

where $\overline{p} = 0.66$, $w(\overline{p}) = 0.57$, $p_3 = 0.83$, and $w(p_3) = 0.83$.

The Bayes' theorem was used to derive the relation $w(p') = \frac{w_1(p) \times L(p)}{\sum w_1(p) \times L(p)}$, where $L(p)$ is the maximum likelihood distribution. The steps involved in deriving the Bayesian probability are as follows:

Step 1: Define the high-probability zone ($p = 0.66$–$1$) and use the conditional probability relation of Bayes' theorem to calculate the Bayesian relation of two connected probabilities. Calculate the Bayesian relation of two connected probabilities and subsequently derive the Bayesian probability distribution through the steps in Table A2.

$$e^{-\beta(-lnp)^{\alpha}} = \frac{e^{-\beta_0}(-lnp)^{\alpha_0} \times L(p)}{\sum_{p=0.66}^{1} e^{-\beta_0}(-lnp)^{\alpha_0} \times L(p)} \tag{16a}$$

$$e^{-\beta(-lnp_2)^{\alpha}} = \frac{e^{-\beta_0}(-lnp_2)^{\alpha_0} \times L(p_2)}{\sum_{p=0.66}^{1} e^{-\beta_0}(-lnp)^{\alpha_0} \times L(p)} \tag{16b}$$

$$e^{-\beta(-lnp_i)^{\alpha}} = \frac{e^{-\beta_0}(-lnp_i)^{\alpha_0} \times L(p_i)}{\sum_{p=0.66}^{1} e^{-\beta_0}(-lnp)^{\alpha_0} \times L(p)} \tag{16c}$$

$$e^{-\beta(-lnp_n)^{\alpha}} = \frac{e^{-\beta_0}(-lnp_n)^{\alpha_0} \times L(p_n)}{\sum_{p=0.66}^{1} e^{-\beta_0}(-lnp)^{\alpha_0} \times L(p)} \tag{16d}$$

Step 2: After dividing the Bayesian relations of the preceding and following terms, let $L(p_1) = 1$ and sum the overall Bayesian relations' values.

$$\frac{(16-2)}{(16-1)} \qquad \frac{e^{-\beta}(-\ln p_2)^\alpha}{e^{-\beta}(-\ln p_1)^\alpha} = \frac{e^{-\beta_0}(-\ln p_2)^{\alpha_0} \times L(p_2)}{e^{-\beta_0}(-\ln p_1)^{\alpha_0} \times L(p_1)} \tag{17a}$$

$$\frac{L(p_2)}{L(p_1)} = \frac{e^{-\beta(-\ln p_2)^\alpha}}{e^{-\beta(-\ln p_1)^\alpha}} \times \frac{e^{-\beta_0(-\ln p_1)^{\alpha_0}}}{e^{-\beta_0(-\ln p_2)^{\alpha_0}}} \tag{17b}$$

$$\frac{L(p_3)}{L(p_2)} = \frac{e^{-\beta(-\ln p_3)^\alpha}}{e^{-\beta(-\ln p_2)^\alpha}} \times \frac{e^{-\beta_0(-\ln p_2)^{\alpha_0}}}{e^{-\beta_0(-\ln p_3)^{\alpha_0}}} \tag{17c}$$

$$\frac{L(p_i)}{L(p_{i-1})} = \frac{e^{-\beta(-\ln p_i)^\alpha}}{e^{-\beta(-\ln p_{i-1})^\alpha}} \times \frac{e^{-\beta_0(-\ln p_{i-1})^{\alpha_0}}}{e^{-\beta_0(-\ln p_i)^{\alpha_0}}} \tag{17d}$$

$$\frac{(16-n)}{(16-n-1)} \qquad \frac{L(p_n)}{L(p_{n-1})} = \frac{e^{-\beta(-\ln p_n)^\alpha}}{e^{-\beta(-\ln p_{n-1})^\alpha}} \times \frac{e^{-\beta_0(-\ln p_{n-1})^{\alpha_0}}}{e^{-\beta_0(-\ln p_n)^{\alpha_0}}} \tag{17e}$$

Step 3: First, let $L(p_1) = 1$, obtain $L(p_2)$–$L(p_n)$ form Equations (17a)–(17e) and sum the values of the overall Bayesian relation. Subsequently, divide the value of $\sum_{i=1}^{n} L(p_i)$ into each formula to obtain the likelihood function $\overline{L}(p_i)$ and calculate the Bayesian probability while satisfying the equation $\sum_{i=1}^{n} L(p_i) = 1$.

Step 4: Using the relation between the high-probability zone ($P = 0.66$–1) and the conditional probability in Bayes' theorem, obtain the Bayesian probability by summing and weighting. As illustrated in the left-hand-side picture in Figure 6, the CDF of the Bayesian probability distribution ($\overline{L}(p_i) = -0.14 \times (p_i - 0.85) \times 2 + 0.07$) approximates a normal distribution ($R^2 = 0.9548$) with a peak at 0.85.

Step 5: As illustrated in the right-hand-side picture in Figure 6, the PDF of the Bayesian probability ($\overline{L}(p_i) = -0.21 \times (p_i - 0.51) \times 2 + 0.05$) distribution also approximates the normal distribution ($R^2 = 0.9999$).

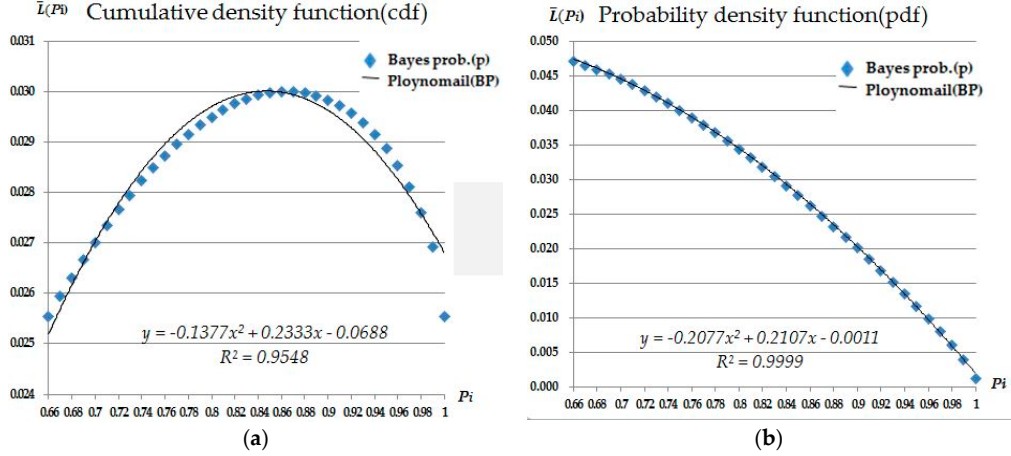

**Figure 6.** Bayesian probability function: (**a**) for cumulative density function; (**b**) for probability density function.

In summary, the high-probability zone of the CCPT approximates the posterior probability of BT. In other words, the provision of external information helps committee members to increase their subjective risk probability when selecting contractors.

## 6. Assessment of the Utility of Bid Commitment

The present research used the FUF developed by Kirkwood [16] and Cheng and Kang [17], which is an extension of this approach, to incorporate the uncertainties of the expert estimates.

Expert questionnaire results were collected and FUFs for D/C/Q commitments were established. Subsequently, the utility of bid commitment for committee members was evaluated by defuzzifying and defining the D/C/Q fuzzy intervals. Thus, the difference between the utilities of influence factors for candidate contractors and the D/C/Q fuzzy weights were determined. Figure 7 illustrates the workflow for evaluating the utility of bid commitment for committee members. The center-of-gravity method was employed to evaluate the difference between the utilities of candidate contractors. After the duration discount, cost discount, and quality assurance data (in %) were converted, the FUF was used to calculate the utility of the bid commitment for committee members.

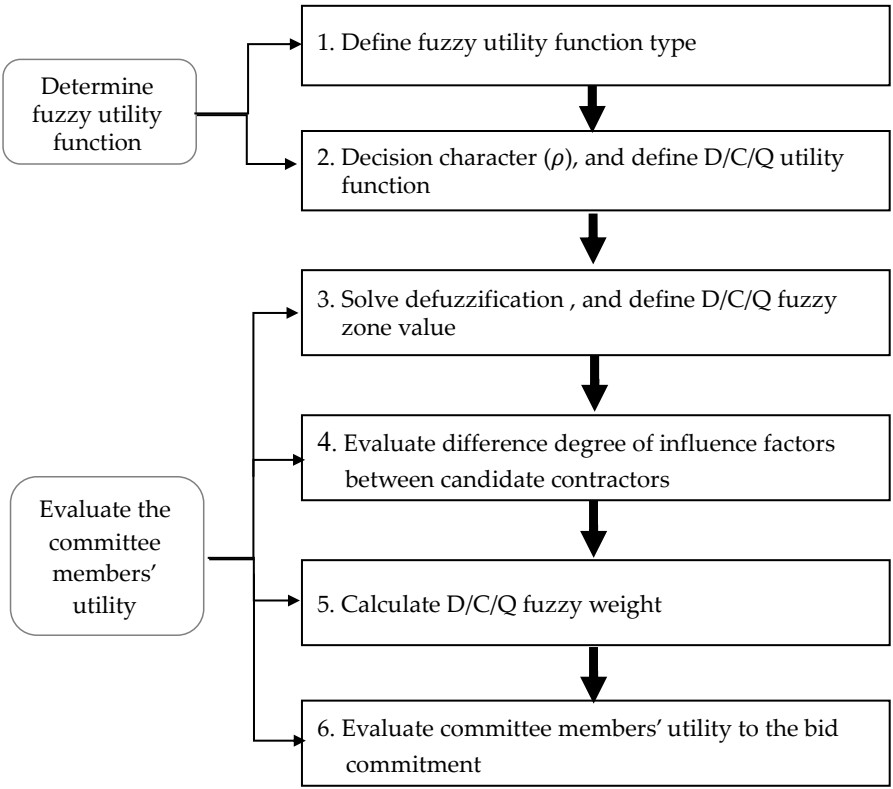

**Figure 7.** Evaluation steps of committee members' utility to the bid commitment.

The D/C/Q utility functions were determined as follows (Figure 8):

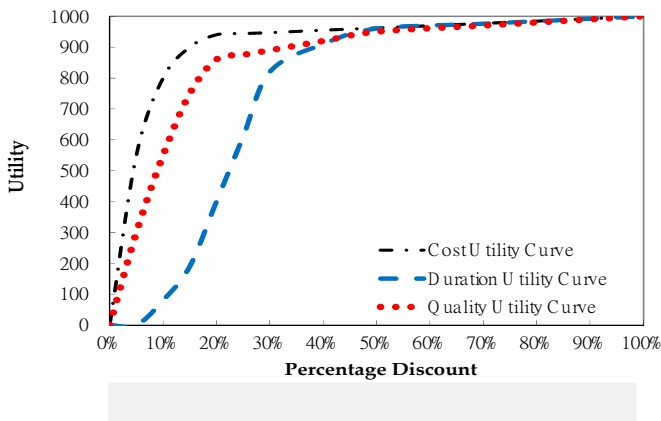

**Figure 8.** D/C/Q utility function.

### 6.1. Determining the Fuzzy Utility Functions

The exponential utility function ($x$) was adopted. In the expression for $\mu(x)$, $x$ represents the preference in decision making, $\rho$ is the risk tolerance of a decision maker, "*Low*" represents the least preferred candidate, "*High*" represents the most preferred candidate, and $\rho$ reflects the personality of a decision maker (conservative, neutral, or adventurous) [16] (p. 6).

(1)  $\rho \geq 0$ Conservative (risk-averse nature)
(2)  $\rho < 0$ Adventurous (risk-seeking nature)

The FUF and shapes are defined Equation (18) as follows:

$$u(x) = \begin{cases} \dfrac{e^{[-\frac{x-Low}{\rho}]}-1}{e^{[-\frac{High-Low}{\rho}]}-1} & ,\rho \neq \text{infinity} \\[12pt] \dfrac{x-Low}{High-Low} & ,\quad \text{others} \end{cases} \tag{18}$$

Specific utility functions, which were established using the data obtained through an expert questionnaire [12] (p. 1057) and the aforementioned information, were used to model the personality of a decision maker ($\rho$). Duration utility function: $u(x) = 1000 \times \dfrac{e^{[-\frac{x-0}{\rho}]}-1}{e^{[-\frac{1-0}{\rho}]}-1}$, where $\rho = 0.2$.

1.  Cost utility function: $u(x) = 1000 \times \dfrac{e^{[-\frac{x-0}{\rho}]}-1}{e^{[-\frac{1-0}{\rho}]}-1}$, where $\rho = 0.06$.

2.  Quality utility function: $u(x) = 1000 \times \dfrac{e^{[-\frac{x-0}{\rho}]}-1}{e^{[-\frac{1-0}{\rho}]}-1}$, where $\rho = 0.12$.

### 6.2. Evaluating a Committee Members' Utility of the Bid Commitment

Three defuzzification methods exist: the center-of-gravity, center of maxima, and center of sums methods. In this study, the center-of-gravity method was adopted to determine the difference degrees of the D/C/Q fuzzy utilities of the candidate contractors. The center of sums method was used to calculate the D/C/Q fuzzy weights to forecast the utility of bid commitment for committee members.

1.  Solve fuzzy weights relative to the D/C/Q factors

The center of sums method was adopted to determine the fuzzy weights between the D/C/Q factors by conducting group decision analysis among the evaluators (Table 9). Subsequently, Equations (19a)–(19g) were used to calculate the ratios of project duration to the cost and the quality fuzzy weights. Equations (19d)–(19f) were used to determine the ranking function of the triangular fuzzy number [63]. The result was ***wd:wc:wq*** = 0.402:0.302:0.296. Thus, from each committee members' perspective, the completion of a project within the assigned duration is the most crucial aspect, followed by meeting the cost and quality requirements. The cost and quality requirements were of equal importance.

$$W_{1j} = \left( \sum W_{ij}, \sum W_{ij}, \sum W_{ij} \right) = (3.3, 3.5, 3.7) \tag{19a}$$

$$W_{2j} = \left( \sum W_{ij}, \sum W_{ij}, \sum W_{ij} \right) = (2.17, 2.6, 3.19) \tag{19b}$$

$$W_{3j} = \left( \sum W_{ij}, \sum W_{ij}, \sum W_{ij} \right) = (2.12, 2.55, 3.15) \tag{19c}$$

$$U\left(W_{1j}\right) = U(3.3, 3.5, 3.7) = \frac{(3.3 + 3.5 \times 2 + 3.7)}{4} = 3.5 \tag{19d}$$

$$U\left(W_{2j}\right) = U(2.17, 2.6, 3.9) = \frac{(2.17 + 2.6 \times 2 + 3.19)}{4} = 2.64 \tag{19e}$$

$$U\left(W_{3j}\right) = U(2.12, 2.55, 3.15) = \frac{(2.12 + 2.55 \times 2 + 3.15)}{4} = 2.59 \tag{19f}$$

$$\boldsymbol{wd} = \frac{U\left(W_{1j}\right)}{\sum U\left(W_{ij}\right)} = \frac{3.5}{3.5 + 2.64 + 2.59} = 0.402; \; \boldsymbol{wc} = 0.302 \; ; \; \boldsymbol{wq} = 0.296 \tag{19g}$$

where *wd*, *wc*, *wq* are weight of the D/C/Q factors; $U\left(W_{1j}\right)$, $U\left(W_{2j}\right)$, $U\left(W_{3j}\right)$ are the triangular fuzzy number of the D/C/Q factors.

**Table 9.** Fuzzy weight value of the D/C/Q factors.

| Factors | Duration | Cost | Quality |
|---|---|---|---|
| Duration (W1j) | (1,1,1) | (1.2,1.3,1.4) | (1.1,1.2,1.3) |
| Cost (W2j) | (0.5,0.6,0.7) | (1,1,1) | (0.67,1,1.49) |
| Quality (W3j) | (0.45,0.55,0.65) | (0.67,1,1.5) | (1,1,1) |

2.  The steps used to determine the difference between the candidate contractors with respect to the D/C/Q fuzzy utility are as follows:

Step 1: Establish a membership function for linguistic variables pertaining to the aforementioned difference.

Fuzzy statistical analysis [64] (pp. 71–72) was used to establish fuzzy ratings for linguistic variables, where the variables were rated in terms of the five levels: very high (VH), high (H), indifference (I), low (L), and very low (VL). Subsequently, fuzzy additive and scalar multiplication methods were used to analyze evaluator responses to questionnaires on the differences for linguistic variables [65] (pp. 231–232). Specifically, the fuzzy numbers representing such difference were calculated according to the method of Cheng and Hsiang [66] to obtain the membership function (Table 10 and Figure 9). Finally, based on the relative frequency (degree of membership function), the statistical linguistic variables were grouped to obtain and correct the fuzzy number of the membership functions.

Step 2: Evaluation of the degree of difference.

The linguistic variables used by evaluators were analyzed to assess the difference between the duration discounts of contractors A and B.

$$\frac{1}{n}(A_1 \oplus \cdots \oplus A_n) = \left[\frac{1}{n} \times (a_1 + \cdots + a_n), \cdots, \frac{1}{n} \times (l_1 + \cdots + l_n)\right] \quad n = 5 \tag{20}$$

where $A_1$–$A_n$ are fuzzy numbers for experts to assess the degree of difference; $a_1$–$a_n$, ... , $l_1$–$l_n$ are the fuzzy number of linguistic variables; *n* is number of experts.

**Table 10.** Fuzzy numbers of semantic variables after linear interpolation.

| Linguistic Variables | The Fuzzy Number (X) | | | | | | | | | | | |
|---|---|---|---|---|---|---|---|---|---|---|---|---|
| | a | b | c | d | e | f | g | h | i | j | k | l |
| VH | 6.50 | 7.50 | 8.50 | 8.88 | 9.50 | 10.00 | 10.00 | 10.00 | 10.00 | 10.00 | 10.00 | 10.00 |
| H | 5.50 | 5.83 | 6.50 | 6.88 | 7.50 | 7.50 | 8.50 | 8.56 | 8.75 | 9.19 | 9.25 | 9.50 |
| I | 3.50 | 3.77 | 4.30 | 4.50 | 5.50 | 5.50 | 6.30 | 6.50 | 6.80 | 7.50 | 7.70 | 8.50 |
| L | 1.50 | 1.77 | 2.30 | 2.50 | 3.50 | 3.50 | 4.30 | 4.50 | 4.80 | 5.50 | 5.70 | 6.50 |
| VL | 0.00 | 0.00 | 0.00 | 0.00 | 0.00 | 1.50 | 2.00 | 2.13 | 2.50 | 3.38 | 3.50 | 4.50 |
| | The Experts Filled in Linguistic Variables [VH, L, VH, I, H] | | | | | | | | | | | |
| SUM | 23.50 | 26.37 | 30.10 | 31.63 | 35.50 | 36.50 | 39.10 | 39.56 | 40.35 | 42.19 | 42.65 | 44.50 |
| Aggregated | 4.70 | 5.27 | 6.02 | 6.33 | 7.10 | 7.30 | 7.82 | 7.91 | 8.07 | 8.44 | 8.53 | 8.90 |

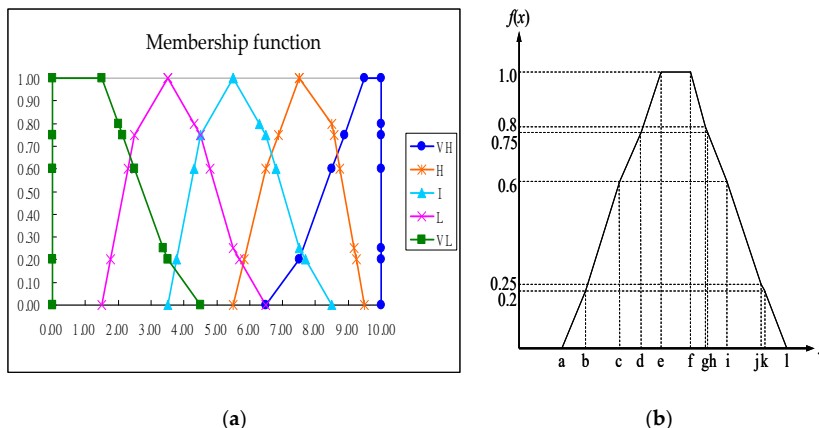

**(a)**                     **(b)**

**Figure 9.** Membership function for fuzzy rating: (**a**) for membership function of linguistic variables after linear interpolation; (**b**) for fuzzy number function of linguistic variables after linear interpolation.

The experts filled in a form of linguistic variables, which are VH, L, VH, I, and H in Table 10. Equation (20) was used to obtain the fuzzy numbers representing the evaluation results of five experts. Specifically, the numbers pertained to the difference between the bidding prices of contractors A and B. Through the calculation $[0.2 \times (6.50 + 1.50 + 6.50 + 3.50 + 5.50), \ldots, 0.2 \times (10.00 + 6.50 + 10.00 + 8.50 + 9.50)]$, the comprehensive fuzzy values were determined to be 4.70, 5.27, 6.02, 6.33, 7.10, 7.30, 7.82, 7.91, 8.07, 8.44, 8.53, and 8.90. The details of the calculations are presented in Figure 10. The center-of-gravity method was used to defuzzify the integrated fuzzy numbers. Using a difference score ($x$) of 6.9571 and a scale of −10 to 10 (which represents a difference of −20% to 20%), a difference ratio of 7.83% was obtained through linear conversion. The degrees of difference in the utilities of duration discount, cost discount, and quality assurance between contractor A and the other contractors were estimated (Table 11).

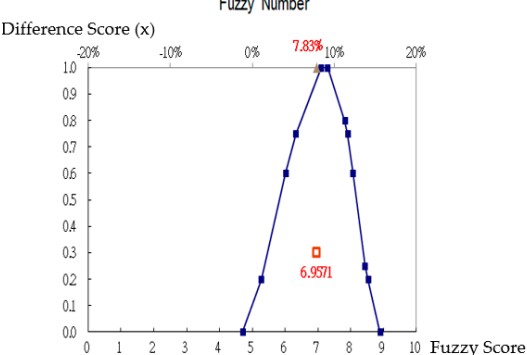

**Figure 10.** Experts' fuzzy linguistic scores for the degree of difference between contractors B and A.

**Table 11.** Difference between contractor A and each contractor with respect to utility.

| Candidate Contractor | Bid Duration (Months) | Bid Cost (TWD) | Duration Discount ($dj$) (%) | Cost Discount ($cj$) (%) | Quality Assurance ($qj$) (%) |
|---|---|---|---|---|---|
| A | 37 | 177,072,000 | - | - | - |
| B | 36 | 173,121,000 | 7.83 | 15.69 | 10.95 |
| C | 40 | 170,884,000 | −5.49 | 15.69 | 5.68 |
| D | 34 | 178,937,000 | 9.43 | −10.43 | 4.16 |
| E | 37 | 175,393,000 | 4.16 | 9.43 | 5.68 |

Step 3: Present the evaluation results of the utility of the bid commitment for the committee members.

The expert questionnaire scores were averaged to determine the duration discount, cost discount, and quality assurance values of contractor A. Subsequently, the difference in utility between each contractor (Table 11) was converted to calculate the duration discount, cost discount, and quality assurance values. The contractor's D/C/Q utilities were also calculated using the data presented in Figure 10. Finally, the ratios of the D/C/Q fuzzy weights, that is, ***wd:wc:wq*** = 0.402:0.302:0.296, were multiplied to obtain the overall utility values, which are listed in Table 12.

**Table 12.** Evaluation results of the committee members' utility to the bid commitment.

| Candidate Contractor | Duration Discount ($d_j$) (%) | Duration Utility (U$d$) | Cost Discount ($c_j$) (%) | Cost Utility (UC) | Quality Assurance ($q_j$) (%) | Quality Utility (U$q$) | Overall Utility Value (TU) |
|---|---|---|---|---|---|---|---|
| A | 26.20 | 735 | 11.30 | 848 | 18.80 | 791 | 785.7 |
| B | 28.25 | 762 | 13.07 | 887 | 20.86 | 824 | 818.1 |
| C | 24.76 | 715 | 13.07 | 887 | 19.87 | 809 | 794.8 |
| D | 28.67 | 767 | 10.12 | 815 | 19.58 | 791 | 788.6 |
| E | 27.29 | 750 | 12.37 | 873 | 19.87 | 791 | 799.3 |

## 7. Evaluation of Overall Prospect Value of Candidate Contractors

This section focuses on CCPT. The posterior probability of candidate contractors (as calculated in Sections 4.3 and 5.2) were multiplied with the utility of bid commitment for committee members (presented in Section 6.2) to obtain the overall prospect value of the candidate contractors. Figure 11 illustrates the evaluation steps. After the contractors were ranked, the best and second-best contractors were selected. Finally, the contractor selection results obtained from BFPM were compared with the lowest tender and overall utility values as well as the MCPM results.

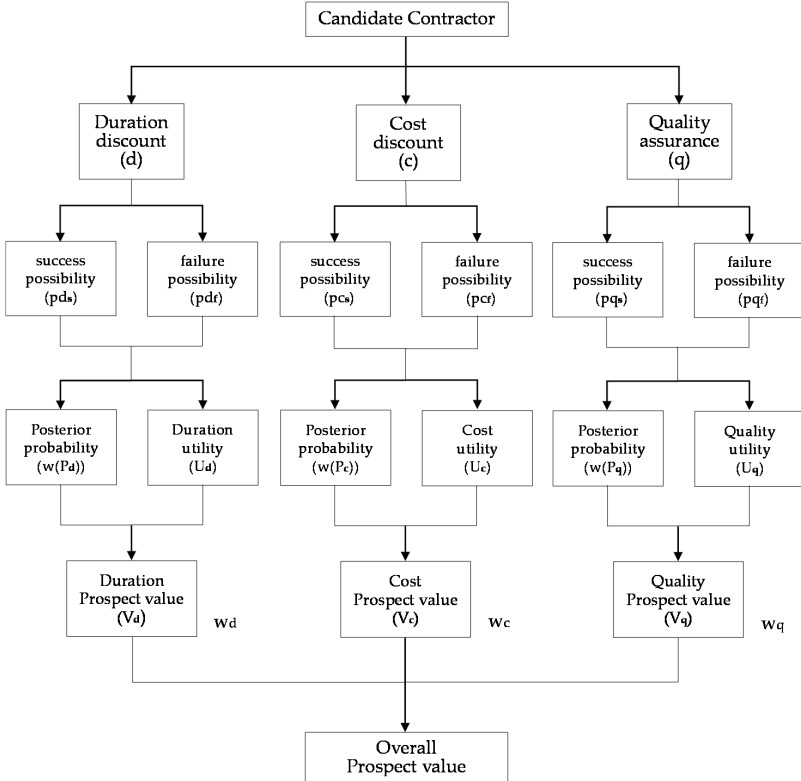

**Figure 11.** Steps for evaluating the overall prospect value of candidate contractors.

### 7.1. Calculation of the Posterior Probability of Candidate Contractors

The implementation possibilities for the bidder's duration, cost, and quality commitments (probability of success) presented in Section 4.3 were converted to the prior probability of duration, cost, and quality factors for committee members through the Bayesian probability weighting functions (Section 5.1). The aforementioned possibilities were also converted to the posterior probability through the Bayesian probability weighting functions (Section 5.2). Subsequently, the posterior probabilities of candidate contractors were summarized to determine the probability of the subjective perception (of risk) of committee members towards a bid contractor. Table 11 lists the prior and posterior probability results.

### 7.2. Evaluation of the Overall Prospect Value of Candidate Contractors

The overall prospect value of a candidate contractor, in terms of the duration, cost, and quality commitments (Table 13), was obtained by multiplying the results listed in Table 10 (utility of bid commitment for committee members) with the results presented in Table 13 (the posterior probability of candidate contractors for committee members). As listed in Table 14, contractor C had the highest prospect for duration, followed by contractors E and A; contractor D had the highest prospect for cost, followed by contractors E and A; and contractor A had the highest prospect for quality, followed by contractors C, D, and E.

**Table 13.** Committee members' posterior probability of candidate contractors.

| Candidate Contractor | Duration | | | Cost | | | Quality | | |
|---|---|---|---|---|---|---|---|---|---|
| | Success Possibility (pds) | Prior Probability (1) | Posterior Probability (2) | Success Possibility (pcs) | Prior Probability (1) | Posterior Probability (2) | Success Possibility (pqs) | Prior Probability (1) | Posterior Probability (2) |
| A | 0.76 | 0.65 | 0.76 | 0.68 | 0.58 | 0.60 | 0.72 | 0.61 | 0.66 |
| B | 0.61 | 0.53 | 0.53 | 0.60 | 0.53 | 0.53 | 0.61 | 0.53 | 0.53 |
| C | 0.83 | 0.71 | 0.83 | 0.50 | 0.46 | 0.46 | 0.67 | 0.58 | 0.60 |
| D | 0.50 | 0.46 | 0.46 | 0.83 | 0.71 | 0.83 | 0.67 | 0.58 | 0.60 |
| E | 0.76 | 0.65 | 0.76 | 0.69 | 0.59 | 0.62 | 0.67 | 0.58 | 0.60 |

**Table 14.** Prospect values for the D/C/Q commitments.

| Candidate Contractor | Duration | | | Cost | | | Quality | | |
|---|---|---|---|---|---|---|---|---|---|
| | Utility $u_d$ | Posterior Probability $w(p_d)$ | Prospect Value $(v_d)$ | Utility $u_c$ | Posterior Probability $w(p_c)$ | Prospect Value $(v_c)$ | Utility $u_q$ | Posterior Probability $w(p_q)$ | Prospect Value $(v_q)$ |
| A | 735 | 0.76 | 558.6 | 848 | 0.60 | 508.8 | 791 | 0.66 | 522.1 |
| B | 762 | 0.53 | 403.9 | 887 | 0.53 | 470.1 | 824 | 0.53 | 436.7 |
| C | 715 | 0.83 | 593.4 | 887 | 0.46 | 408.0 | 887 | 0.59 | 477.3 |
| D | 767 | 0.46 | 352.8 | 815 | 0.83 | 676.4 | 815 | 0.59 | 466.7 |
| E | 750 | 0.76 | 570.0 | 873 | 0.62 | 541.3 | 873 | 0.59 | 466.7 |

The D/C/Q analysis revealed that contractor D provided the largest duration discount. However, as the evaluators perceived contractor D to be least likely to implement its commitments, contractor D had the smallest expected utility value. The opposite was the case for contractor C. Specifically, contractor C provided the smallest duration discount. However, as the evaluators perceived contractor C to be highly likely to implement its commitments, contractor C had the highest expected utility value. Furthermore, contractor C's projected cost was also within the project's budget. The quality commitment of contractor A also had the highest expected utility value, which indicated that contractor A was the most likely to successfully implement the commitment.

### 7.3. Selection of the Optimal Contractor

The aforementioned expected utility analysis was also applied to evaluate the cost discount in the bid commitment. The bid commitments of the five candidate contractors were represented in terms of the overall weighted duration discount, *d*, cost discount, *c*, and quality, *q*. Their corresponding expected utility values, *vd, vc,* and *vq*, are presented in Table 15. The overall prospect utility value was calculated using the following equation:

$$v = wd \times vd + wc \times vc + wq \times vq \tag{21}$$

where *v* is prospect value; *wd*, *wc*, and *wq* are the duration, cost, and quality weights (values are 0.402, 0.302, and 0.296); *vd*, *vc*, and *vq* are expected utility values. As indicated in Table 15, the top three overall prospect values based on the posterior probability obtained from the BFPM were higher than 500. In other words, the BFPM can be used as a contractor selection model for committee members to define the minimum threshold when selecting contractors in the considered project of mass-rapid-transit station development.

**Table 15.** Evaluation results of the BFPM and other decision-making models.

| Candidate Contractor | The Lowest Bid | | MCDM | | MCPM | | BFPM | |
|---|---|---|---|---|---|---|---|---|
| | | | Overall Utility | | Overall Prospect Value (Prior Probability) $(v'_T)$ | | Overall Prospect Value (Posterior Probability) $(v''_T)$ | |
| | Cost Discount | Rank | Utility Value | Rank | Prospect Value | Rank | Prospect Value | Rank |
| A | 11.30% | 4 | 785.7 | 5 | 483.4 | 2 | 532.8 | 1 |
| B | 13.07% | 1 | 818.1 | 1 | 433.6 | 5 | 433.6 | 5 |
| C | 13.07% | 1 | 794.8 | 3 | 466.2 | 3 | 503.0 | 3 |
| D | 10.12% | 5 | 788.6 | 4 | 452.4 | 4 | 484.2 | 4 |
| E | 12.37% | 3 | 799.3 | 2 | 487.3 | 1 | 530.8 | 2 |
| Gap $(\frac{max-min}{min}\%)$ | 2.95% | | 32.4 (4.12%) | | 53.7 (12.38%) | | 99.2 (22.87%) | |
| $(\frac{max-third}{max-min}\%)$ | - | | 71.9% | | 39.3% | | 30.04% | |

When six evaluators were considered in the proposed prediction model, contractor A attained the highest ranking, with the highest prospect utility value and received the highest evaluation. Moreover, contractor A also ranked first in quality, and its expected utility with respect to duration and cost was also above average. Therefore, contractor A was selected as the optimal applicant, followed by contractor E. Contractors B and C provided the lowest cost discount. As the probability of implementing a cost discount was very low, the prospect utility of cost commitment was very low. By contrast, contractor C's duration commitment was positively evaluated by the panel of experts. Therefore, contractor C ranked third with respect to the overall prospect utility. Although contractor B offered the highest duration discount, this contractor performed unfavorably in the possibility of implementing cost commitment. Thus, contractor B had the lowest expected utility. The expected utility of contractor B with respect to quality commitment was low. As the cost discount of contractor B was low, the implementation probability of cost commitment was also low. Therefore, contractor B had the lowest prospect utility, which indicates that the implementation of contractor B's bid commitment carries a high risk.

The results obtained from BFPM and from other contractor selection decision-making models were compared. If the risk in the (unsuccessful) implementation of bid commitment was not considered, the model providing the lowest tender or overall utility value [4] was adopted. Without this consideration of risk, contractor B won the bid and had the highest ranking (Table 14). When the successful implementation of bid commitment considered in the MCPM and BFPM was converted into risk probability, contractors A, C, and E were the top three contractors with the highest overall prospect

values. The duration commitment of contractor C was recognized by the group of evaluators; thus, contractor C ranked third with respect to overall prospect utility. Although the discounts offered by contractors A and E did not match the highest discounts offered by contractors C and D, contractors C and D were less likely to implement their projects successfully. Therefore, both contractors had similar overall evaluation results. In BFPM, the provision of external information was incorporated into the risk probability. As contractor A had the highest prospect utility with respect to quality commitment, this contractor won the bid, followed by contractor E. In general, the contractor offering the highest overall utility may also carry a high risk of poor implementation, relative to the bidder's offers. This phenomenon can be quantitatively represented using the Bayesian posterior probability.

The use of FUT and the Bayesian posterior probability can increase the difference between the overall prospect values, which is in line with the idea of establishing minimum threshold selection criteria for committee members. This approach can be integrated in the selection criteria when releasing the tender documentation announcement. BFPM and MCPM can effectively filter out the options included in contractor selection criteria which carry a high cost risk, despite a high cost utility. Moreover, BFPM and MCPM do not cause malicious competition in the bidding process. In the BFPM, the posterior probability of external information in BT can increase the difference between the ratings of bidding contractors and truly reflect the requirements of establishing a minimum score threshold when public bidding involves procurement procedures. The results of BFPM method can avoid the lowest bidder being selected; in addition, the score gap of contractor selection can be increased from 4.12% (MCDM) to 22.87%, and the difference between the top three scores can be shortened from 71.91% (MCDM) to 30.04%.

## 8. Conclusions

In Taiwan, the most advantageous tender in governmental procurement is the selection of a general contractor based on score or ranking evaluated by committee. In fact, due to personal subjective preference, contractor selection of committee members may be different, causing cognitive difference between the results of the members' selection and the preliminary opinions provided by the working group. Thus, if the overall performance of bid contractors can be predicted before contractor selection, specifically with respect to duration, cost, and quality, the committee can better select the optimal general contractor for the construction project.

In this study, to solve the aforementioned contractor selection problem, we developed a new model, BFPM, which considers the three influence factors of duration, cost, and quality. Four main set evaluation methods were defined. The proposed method of integrating the probability by using Bayes' theorem was used to calculate the subjective utility of factors, which was assessed by using the fuzzy utility, before obtaining the prospect values for contractors multiplied their Bayesian probabilities by utilities. To obtain the overall estimate, fuzzy theory was also used to recalculate the objective weights and combine the subjective risk preference and objective utilities.

The obtained result provides a theoretical basis for using a method with wide practical applications for combining factor weights (obtained by using MPR and FPR methods) as the arithmetic mean of some sets of values. The considered methods use the posterior probability of BT to represent the probability of implementing bid commitment. This model aids committee members in their selection of the best contractor for public construction projects. The results of this study can avoid the lowest bidder being selected; in addition, the score gap of contractor selection can be increased, and the difference between the top three contractors' scores can be decreased. This method further verifies that the committee members establish the minimum threshold criteria of contractor selection. The results of this study not only ensure the duration and cost in public construction, but also promotes project quality. Moreover, our contributions aid sustainability in the operations and development of public infrastructure.

The proposed method combines the risk preference to calculate the factor weights and utilities after obtaining the estimates of a group of experts. The method also transforms subjective preference

into objective weights and utilities. The calculated weights of the factors, the Bayesian probability, and the utility functions can serve as a reference example. This research focuses on the process of selecting optimal contractors, discussing the personal preferences of committee members, and analyzing the members' preference behaviors for contractors through a mathematical model (Bayes theorem et al.) Using the mathematical model, in addition to proposing an innovative decision-making system of contractor selection and an index weight-assessing system for sustainable development, this model will be widely applied and sustainably updated for other similar cases, such as railway station development, urban renewal or social housing buildings of contractor selection for public construction projects. The results of BFPM help to select the best contractor, can be applied to the life cycle construction, and can promote sustainable development.

**Author Contributions:** M.-Y.C. conceived of the main research idea, provided extensive advice throughout the study, and made methodological revisions; S.-H.Y. and W.-C.C. collected the data, conducted the expert interviews, performed calculations, and analyzed the data and results; S.-H.Y. administrated the project and wrote the paper; and M.-Y.C. and S.-H.Y. discussed the model evaluation results and commented on the paper. All authors have read and agreed to the published version of the manuscript.

**Funding:** This study received no external funding.

**Conflicts of Interest:** The authors declare no conflicts of interest.

**Appendix A**

**Table A1.** Aggregation of influence factors for contractor selection at domestic and abroad literatures.

| Factors | Weber (1991) [52] | Dickson (1966) [53] | Choi (1996) [54] | Hsu et al. (1998–2012) [55] | Alzober 2014 [56] | Ebrahi-Mi 2016 [57] | Oyatoye 2016 [58] | Chiang 2017 [59] | Hasnain 2018 [27] | Turskis 2019 [20] | Morkunaite 2019 [26] | Morkunaite 2019 [60] | Maha-Madu 2020 [10] | Koc 2020 [35] | Zhang 2020 [36] | Adoption Factors |
|---|---|---|---|---|---|---|---|---|---|---|---|---|---|---|---|---|
| 1. Plan Management (DF4) | √ | √ | √ | (7) | √ | √ | √ |  | √ |  |  | √ | √ | √ | √ | 18 |
| 2. Green Building |  |  |  | (2) |  |  |  |  |  |  |  |  |  | √ | √ | 4 |
| 3. Building Materials (Capacity)/Equipment Resources QF1) | √ | √ | √ | (3) | √ | √ | √ |  | √ | √ | √ |  | √ | √ | √ | 19 |
| 4. Comfort and Environment |  |  |  | (2) | √ |  |  |  |  |  |  |  |  |  | √ | 4 |
| 5. Migration Compensation |  |  |  | (1) |  |  |  |  |  |  |  |  |  |  |  | 1 |
| 6. Land Equity Conversion | √ |  |  | (1) |  |  |  |  |  |  |  |  |  |  | √ | 3 |
| 7. Trust Management |  |  |  | (1) |  |  |  |  |  |  |  |  |  |  |  | 1 |
| 8. Contract Execution Volume (CF1) | √ | √ | √ | (3) | √ | √ | √ |  | √ | √ |  | √ |  | √ | √ | 18 |
| 9. Goodwill and Industry's Greatest Position (CF2) | √ | √ | √ | (1) | √ |  | √ | √ |  |  | √ | √ | √ |  | √ | 10 |
| 10. Financial Status (Capacity) (DF5) | √ | √ | √ | (6) | √ |  | √ | √ | √ | √ | √ | √ |  | √ | √ | 17 |
| 11. Historical Performance (CF3) | √ | √ | √ | (4) | √ | √ | √ | √ | √ | √ | √ | √ | √ | √ | √ | 16 |
| 12. After-Sales Service (Service Attitude) (QF2) | √ | √ | √ | (3) |  | √ | √ | √ | √ | √ |  |  | √ | √ |  | 12 |
| 13. Warranty Period (QF3) | √ | √ |  | (2) |  | √ | √ |  |  |  |  |  |  | √ | √ | 8 |
| 14. Communication and Coordination with Residents |  |  | √ | (1) |  |  |  |  |  |  |  |  |  |  |  | 2 |

**Table A1.** *Cont*.

| | | | | | | | | | | | | | | | | Total |
|---|---|---|---|---|---|---|---|---|---|---|---|---|---|---|---|---|
| 15. Construction Period or Delivery Capacity (DF6) | √ | √ | √ | (5) | √ | √ | √ | | √ | √ | √ | | √ | √ | √ | 17 |
| 16. Price (Cost) (CF4) | √ | √ | √ | (5) | √ | | √ | √ | √ | √ | √ | √ | √ | √ | √ | 18 |
| 17. Technical Ability (DF1) | √ | √ | | (6) | √ | √ | √ | √ | √ | √ | √ | | √ | √ | √ | 18 |
| 18. Management Organization (Control)(QF4) | √ | | | | √ | √ | | √ | √ | √ | √ | √ | √ | √ | √ | 11 |
| 19. Communication Cooperation/ Subcontracting Situation (QF5) | √ | √ | √ | (3) | √ | | | √ | | √ | √ | √ | √ | √ | | 13 |
| 20. Manufacturer Qualification Manpower (DF2) | √ | √ | √ | (2) | √ | √ | √ | √ | | √ | √ | | √ | √ | | 13 |
| 21 Distance (Location) | √ | √ | | (1) | | | | | | | | | | | | 3 |
| 22. Customer Complaint Procedure | √ | √ | | | | | | | | | | | | √ | | 3 |
| 23. Past Impressions | √ | √ | √ | | √ | | | | | | | | | | √ | 5 |
| 24. Labor Relations (Resolving Conflicts) (DF7) | √ | √ | √ | (1) | √ | | | √ | | | √ | | √ | | | 8 |
| 25. Planning and Control (DF3) | √ | √ | √ | (2) | √ | √ | | | √ | √ | | √ | √ | √ | | 12 |
| 26. Performance Record/Project Claim | | | | (2) | √ | | | | | | | | | √ | √ | 5 |
| 27. Failed Projects | | | | (1) | | | | √ | | | | | | | | 2 |
| 28. Training/Security Management Capabilities | | | | (3) | | √ | √ | | | | | | | | | 5 |
| 29. Creation/Development Potential | | | | | | √ | | | | √ | | √ | | √ | √ | 5 |
| 30. Health and Safety | | | | | √ | | | √ | √ | | √ | | | | √ | 5 |
| Number of Items | 19 | 17 | 14 | 6–15 | 18 | 13 | 13 | 12 | 12 | 13 | 12 | 10 | 13 | 18 | 18 | |

# Appendix B

**Table A2.** Derivation of the Bayes probability weight function.

| p | Cumulative Density Function (CDF) | | | Probability Density Function (PDF) | | | Bayes Relation | | Bayes Probability |
|---|---|---|---|---|---|---|---|---|---|
| | w(p) | w1(p) | B(p) Weight Effect | w(p) | w1(p) | B(p) Weight Effect | | | |
| 0.66 | 0.5697 | 0.5697 | 1.0001 | 0.0072 | 0.0161 | 2.2354 | 0.9893 | 1 | 0.0473 |
| 0.67 | 0.5769 | 0.5858 | 1.0153 | 0.0073 | 0.0161 | 2.2086 | 0.9880 | 0.9880 | 0.0467 |
| 0.68 | 0.5843 | 0.6019 | 1.0300 | 0.0074 | 0.0161 | 2.1790 | 0.9866 | 0.9747 | 0.0461 |
| 0.69 | 0.5918 | 0.6179 | 1.0441 | 0.0075 | 0.0160 | 2.1467 | 0.9852 | 0.9603 | 0.0454 |
| 0.7 | 0.5994 | 0.6339 | 1.0576 | 0.0076 | 0.0160 | 2.1117 | 0.9837 | 0.9447 | 0.0447 |
| 0.71 | 0.6070 | 0.6498 | 1.0704 | 0.0077 | 0.0159 | 2.0743 | 0.9822 | 0.9279 | 0.0439 |
| 0.72 | 0.6148 | 0.6656 | 1.0826 | 0.0078 | 0.0158 | 2.0343 | 0.9807 | 0.9100 | 0.0430 |
| 0.73 | 0.6227 | 0.6814 | 1.0942 | 0.0079 | 0.0157 | 1.9918 | 0.9791 | 0.8910 | 0.0421 |
| 0.74 | 0.6307 | 0.6970 | 1.1050 | 0.0080 | 0.0156 | 1.9470 | 0.9775 | 0.8710 | 0.0412 |
| 0.75 | 0.6389 | 0.7125 | 1.1152 | 0.0082 | 0.0155 | 1.8998 | 0.9758 | 0.8499 | 0.0402 |
| 0.76 | 0.6472 | 0.7278 | 1.1246 | 0.0083 | 0.0154 | 1.8504 | 0.9740 | 0.8277 | 0.0391 |
| 0.77 | 0.6556 | 0.7430 | 1.1333 | 0.0084 | 0.0152 | 1.7987 | 0.9721 | 0.8046 | 0.0381 |
| 0.78 | 0.6642 | 0.7580 | 1.1412 | 0.0086 | 0.0150 | 1.7450 | 0.9701 | 0.7806 | 0.0369 |
| 0.79 | 0.6730 | 0.7729 | 1.1484 | 0.0088 | 0.0148 | 1.6892 | 0.9680 | 0.7556 | 0.0357 |
| 0.8 | 0.6820 | 0.7875 | 1.1547 | 0.0090 | 0.0146 | 1.6313 | 0.9658 | 0.7298 | 0.0345 |
| 0.81 | 0.6912 | 0.8019 | 1.1602 | 0.0092 | 0.0144 | 1.5715 | 0.9634 | 0.7030 | 0.0332 |
| 0.82 | 0.7005 | 0.8161 | 1.1649 | 0.0094 | 0.0142 | 1.5099 | 0.9608 | 0.6754 | 0.0319 |
| 0.83 | 0.7102 | 0.8300 | 1.1687 | 0.0096 | 0.0139 | 1.4464 | 0.9580 | 0.6470 | 0.0306 |
| 0.84 | 0.7200 | 0.8436 | 1.1716 | 0.0099 | 0.0136 | 1.3812 | 0.9549 | 0.6178 | 0.0292 |
| 0.85 | 0.7302 | 0.8570 | 1.1736 | 0.0102 | 0.0133 | 1.3142 | 0.9515 | 0.5879 | 0.0278 |
| 0.86 | 0.7407 | 0.8700 | 1.1746 | 0.0105 | 0.0130 | 1.2456 | 0.9478 | 0.5572 | 0.0264 |
| 0.87 | 0.7515 | 0.8827 | 1.1747 | 0.0108 | 0.0127 | 1.1753 | 0.9436 | 0.5258 | 0.0249 |
| 0.88 | 0.7627 | 0.8951 | 1.1736 | 0.0112 | 0.0123 | 1.1035 | 0.9389 | 0.4936 | 0.0233 |
| 0.89 | 0.7743 | 0.9070 | 1.1715 | 0.0116 | 0.0120 | 1.0301 | 0.9335 | 0.4608 | 0.0218 |
| 0.9 | 0.7864 | 0.9186 | 1.1681 | 0.0121 | 0.0116 | 0.9552 | 0.9273 | 0.4273 | 0.0202 |
| 0.91 | 0.7990 | 0.9297 | 1.1636 | 0.0126 | 0.0111 | 0.8787 | 0.9199 | 0.3931 | 0.0186 |
| 0.92 | 0.8123 | 0.9403 | 1.1576 | 0.0133 | 0.0106 | 0.8007 | 0.9112 | 0.3582 | 0.0169 |
| 0.93 | 0.8263 | 0.9505 | 1.1502 | 0.0141 | 0.0101 | 0.7210 | 0.9005 | 0.3225 | 0.0153 |
| 0.94 | 0.8413 | 0.9600 | 1.1411 | 0.0150 | 0.0096 | 0.6396 | 0.8870 | 0.2861 | 0.0135 |
| 0.95 | 0.8574 | 0.9690 | 1.1301 | 0.0161 | 0.0090 | 0.5562 | 0.8696 | 0.2488 | 0.0118 |
| 0.96 | 0.8750 | 0.9773 | 1.1169 | 0.0176 | 0.0083 | 0.4705 | 0.8459 | 0.2105 | 0.0100 |
| 0.97 | 0.8946 | 0.9848 | 1.1007 | 0.0196 | 0.0075 | 0.3819 | 0.8117 | 0.1708 | 0.0081 |
| 0.98 | 0.9173 | 0.9913 | 1.0807 | 0.0227 | 0.0066 | 0.2893 | 0.7574 | 0.1294 | 0.0061 |
| 0.99 | 0.9455 | 0.9967 | 1.0541 | 0.0282 | 0.0054 | 0.1897 | 0.6557 | 0.0848 | 0.0040 |
| 1 | 1.0000 | 1.0000 | 1.0000 | 0.0545 | 0.0033 | 0.0611 | 0.3222 | 0.0273 | 0.0013 |
| Overall | | | | 0.4082 | 0.3821 | | | 21.1433 | 1.0000 |

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
