# Peer review of "Multi-Criteria Decision Making of Contractor Selection in Mass Rapid Transit Station Development Using Bayesian Fuzzy Prospect Model"

_sustainability, doi:10.3390/su12114606_

Round 1

Reviewer 1 Report

Highlight changes in yellow in a next revision, please. No track changes.

Consider comments in the entire text.

I believe the language used in this text, despite having been improved in terms of English, needs to see improvements in the scientific terms to be used, and in clarity and assertiveness.

Just as example: “In Taiwan, the most advantageous tender of the government procurement is to select the 13 general contractor, based on the score or ranking evaluated by the committee.”

The language used is poor, ant it compromises the entire text, I am sorry.

“Practically, 14 committee members may have divergent selections, and contract selection methods also may be 15 inconsistent in the bid documents causing disputes and appeals of bid contractors on the 16 procurement process and results.”

Language is incorrect, do not refer to the study like this: “This study developed the Bayesian”

I will make general comments since the language does not allow to properly address eth issues.:

Please entirely scientifically  revise the text in terms of language

The structure of the abstract will reflect the quality of the entire text: authors end with a reference to methodology… to be entirely revised…

“This study developed the Bayesian”:

As in conclusions:

Brief contextualization

Brief methodology

Main findings

Practical implications…

See that the language used is not enlightening, it is not clear and does not properly address the context... “a decision-maker can rank 32 alternatives and select the best one.”

Where is (CPT) defined earlier? “PT (CPT) [14],”

There are many abbreviations which were never defined, as example: “AHP,”

Why the “;” The reference style used is not correct and is misleading…

 “Revie, M. ; Bedford, T. [12]”

References must not come after “.” I can see that changes were made which were not indicated in yellow. “event. [45,46].”

Authors must entirely revise mathematical nomenclature, as italics…

Example: “p(?|?) = ??????∙?(?)

?(?) , (1)” equation 1, it is a matter of coherence…

Also, in the text…

For a mathematical text to be submitted to a s «journal such as sustainability, there must be a clear connection to the journal, rather that the mathematical methods being exposed…

I believe that is not the case here…

This short summary subsection in Literature review is not highlighting to me. The way authors express statements do not link the results from literature to a journal like sustainability

So, it is in fact a procedure, better aimed to other journals’ scopes… “This paper presents a decision-making procedure for selecting a general contractor for 170 construction projects.”

Authors do return to literature again here: “4. Assessment Implementation Possibility for Bid Commitment

226 4.1. Identifying Influence Factors of Duration, Cost, and Quality Implementation

227 A review of the literature on the following topics was conducted”

And the structure in the text seems now to want to justify particular aspects from the text, which, however, always focus on the mathematical aspects of models being addressed.

In my perspective, besides better addressing the logical structure of this text, authors would need to convert the text into a relevant text in sustainability area.

Involving questionnaires, where is ethical committees’ information and informed consent?

HUGE issue, in my opinion.

Use assessment instead: “Evaluation”

In Table 12, why use “%” everywhere next to every value

Please connect final statements to a journal with the scope of sustainability… “obtaining the estimates of groups of experts. The method also transforms subjective

635 preference into objective weights and utilities. The calculated weights of the factors, the Bayesian 636 probability, and the utility functions can be a reference example and applied to the 637 decision-making of contractor selection in domains such as railway station development or social 638 housing construction.”

As described, the procedures described thought the etc do not address sustainability issues, within the scope of Sustainability Journal.

References: not a single reference from 2020 is cited…

References 20, 26, 38, 46 refer to Sustainability Journal

I believe some of them better address the scope

All equations should be numbered and properly addressed in the text

Reviewer 2 Report

The authors took into account the comments of the reviewers, made a lot of corrections and greatly improved the quality of the article.

The article may be published after a minor revision.

There is a lot of obscurity in the references list. Reviewer present a revised references:

22 Should be: Keshavarz-Ghorabaee, M.; Amiri, M.; Zavadskas, E. K.; Turskis, Z.; Antucheviciene, J. A Dynamic Fuzzy Approach Based on the EDAS Method for Multi-Criteria Subcontractor Evaluation. Information. 2018, 9(3), 68.

23 Should be: Davoudabadi, R.; Mousavi, S.M.; Saparauskas, J. ; Gitinavard ,H. Solving construction project selection problem by a new uncertain weighting and ranking based on compromise solution with linear assignment approach. Journal of Civil Engineering and Management. 2019, 25(3), 241-251.

24  Should be: Jato-Espino, D.; Castillo-Lopez, E.; Rodriguez-Hernandez, J.; Canteras-Jordana, J.C. A review of application of multi-criteria decision making methods in construction. Automation in construction. 2014, 45, 151-152.

25 Should be: Ilce, A.C.; Ozkaya K. An integrated intelligent system for construction industry: A case study of raised floor material. Technological and Economic Development of Economy. 2018, 24(5), 1866-1884.

29 Should be: Khanzadi, M.; Turskis, Z.; Amiri, G.G.; Chalekaee, A. A model of discrete zero-sum two-person matrix games with grey numbers to solve dispute resolution problems in construction. Journal of Civil Engineering and Management. 2017, 23(6), 824-835.

31 Should be: Morkunaite, Z.; Podvezko, V.; Zavadskas, E.K.; Bausys, R. Contractor selection for renovation of cultural heritage buildings by PROMETHEE method. Archives of Civil and Mechanical Engineering.2019, 19(4), 1056-1071.

32  Should be: Gunduz, M.; Alfar, M. Integration of Innovation through Analytical Hierarchy Process (AHP) in Project Management and Planning. Technological and Economic Development of Economy. 2019, 25(2), 258-276.

35 Should be: Ye, K.H.; Zeng, D.; Wong, J. Competition rule of the multi-criteria approach: what contractors in China really want?. Journal of Civil Engineering and Management. 2018, 24(2), 155-166.

36 Should be: Ortiz, O.I.; Pellicer, E.; Molenaar, K.R. Management of time and cost contingencies in construction

projects: a contractor perspective. Journal of Civil Engineering and Management. 2018, 24(3), 254-264.

46 Should be: Le´sniak, A.; Janowiec, F. Risk Assessment of Additional Works in Railway Construction Investments Using the Bayes Network. Sustainability. 2019, 11, 5388, 1-15.

60 Should be: Morkunaite, Z.; Podvezko, V. Criteria Evaluation for Contractor Selection in Cultural Heritage Projects Using Multiple Criteria Approach. In proceedings of the 17th International Colloquium on Sustainable Decisions in Built Environment, Lithuania Vilnius Gediminas Technical University, 15 May 2019.

Round 2

Reviewer 1 Report

Highlight changes in yellow in a next revision, please. No track changes.

Consider comments in the entire text.

Again, mathematical nomenclature is not coherent, please see mathematical expressions, either using italics and not…

Each parameter (and included units inside (), where available) should be defined after EACH equation…

Even the size differs, important in an internationally indexed Journal…

Also in Figures, as Figure 6, for example, with gray shadows ad mathematical information without italics…

[italics issue is reflected in the entire text, it compromises the coherence to exist in mathematical nomenclature]

Figure 6 as example: two figures are included. Then, after the main caption, a separate detailed caption must be presented for each a) and b) figures, as usual…

Figure 1: all text is distorted…

[also in Figure 3…]

Text in front of lines… ☹

I can see a table next to Figure 10 (where axis have no legends)

Appendix information is supplementary information, it should not be directly addressed in the text...

“From Appendix A,”

Unfortunately, English remains poor, as seen: “There are 11 influence factors belong to sustainable development 236 indexes,”

Well, this is a serios issue, because ethics must be involved each time human information is obtained… always… before he study takes place

Response 5:

The authors have drawn up the statement that the experts agreed to make the questionnaire contentes public in the beginning of questionnaire. In addition, I consulted the staff at our university and she said that we had currently "National Taiwan University of Science and Technology Academic ethics management and self-regulation" as a rule, there is no ethics committee information and informed consent.”

Additional information would have to be included in the text.

You may include as many “Sustainability” references as you want, but in YOUR paper, sustainability must be properly addressed, it has improved, but not enough

Author Response

 Response to Reviewer 1 Comments (2)

This manuscript is a resubmission of an earlier submission. The following is a list of the peer review reports and author responses from that submission.

Round 1

Reviewer 1 Report

General comments to the article:

I found article very interesting. The research design is appropriate. The title of the paper reflects its content. The structure of the article is very well prepared. The figures are necessary. Especially Figure 1. Constructing the BFPM is very useful for readers. The subject matter is within the scope of the Journal.

The reviewer suggestions:

The abstract must be improved. It should have a total of about 200 words maximum. According the instructions for Authors it should consist 1) Background: Place the question addressed in a broad context and highlight the purpose of the study; 2) Methods: Describe briefly the main methods or treatments applied. 3) Results: Summarize the article's main findings; and 4) Conclusion: Indicate the main conclusions or interpretations. In Authors’ Abstract I have founded the point 2)  In general, literature review is purposefully oriented, but it should be supplemented by several newest references. In list of references I have only found 4 papers above 2010. It's only 13%, and most of them are older, some from 1973! I am sure that in this area of research the literature studies can be refreshed a bit. I suggest reference the current publications offer a decision model of contractor selection also. In chapter 4.1 cite research related to the investigation and the relative importance of the contractor selection criteria  and the key determinants of contractor performance and relative importance of quality assessment. Why do authors use factors identified 10 years ago and more? Are they still current? Does the choice of contractor still is based on the same criteria? How  economic changes, changing legal regulations or the type of country influence on it.?  This must be explained I suggest checking formulas numbering - Formula 6 is missing. I suggest the authors write about the pros and cons of the proposed methods. Is it better than other methods of contractor selection proposed in the literature? If yes… why? Please improved the summary chapter.

Reviewer 2 Report

Highlight changes in yellow in a next revision, please. No track changes.

Consider comments in the entire text.

Please remove all underline from abstract.

AS seen, the entire text will need overall proofreading: “In this study, in terms of the sustainability of the construction life cycle, a multi-criteria”

All abbreviations must be defined at first use, abstract and text… “MCDM” check all.

See this is unacceptable English: “According to external information provided with committee members can effectively assess the implementation possibility”

Not evaluation but assessment.  “Finally, the optimal applicant can be selected according to evaluation results.”

The entire scientific language must be revised…

See that abstract usually addresses:

Brief contextualization

Brief methodology

Main findings

Practical implications

Clear results must be presented

There must not be such a number of Keywords. Please see: https://www.mdpi.com/journal/sustainability/instructions

Not even a single reference, never seen this: “1. Introduction

Oh! They are scarce and authors did decide to use a different style, not possible… “(Cheng, Hsing, and Chuang, 2011).”

All references must be revised entirely... “Topcu (2004) [1]”

Authors cannot use mixed styles

Address all correct spacing in the whole document: “2.2. Preference Relationships Theory(PR)

[In many cases… “(pp.456-460)employed”]

See that this is literature review. The reference to methodology I the study, should be briefly referred at the end of literature review

2.2. Preference Relationships Theory(PR)

115 This study was based on the preference”

This type of statements should not be “scattered” through the text, but together…

3. Constructing a Bayes Fuzzy Prospect Model

182 This study aimed to develop”

Please address all italics and parameters and bold in matrix/vector data

Please clarify spacing and abbreviations: “NT$199”

All captions must be self-explanatory. And avoid abbreviations:

Figure 1. Constructing the BFPM”

Incorrect English: “4. Implementation Possibility Evaluation of Bid Commitment

Non-acceptable English: “Figure 2. Relative important MPR matrix of duration 274 discount factor”

Isi is very difficult to understand some terms, used in improper context: “For illustration, a(df)17”

Assure that all known data (equations, mathematical data) are indicated along with the source citations, otherwise highlight the originality and novelty in each case, to be clear what is newly being presented.

A succession of mathematical data um a journal like Sustainability must clearly link to the sustainability issues… Otherwise, a mathematical journal should be chosen

See that the quality of figures lacks to me improved, and be consistent through the text…

Authors continue to mix reference style… “Ali al-Nowaihi & Sanjit Dhami (2010)”

All issues must be addressed before a submission..

How do ayhors link presented mathematical data to text…?

“The posterior probability of Bayes’ theorem

377 was set as the risk probability, as Figure 5 showed.”

Figure 6 lacks proper axis information… avoid red text and there is unreadable content (xx axis…

So many things to be addressed

An extensive number of major headings do not contribute to paper clarity…

6. Utility Evaluation of Bid Commitment

It almost seems the first reference to the study… “In this study, the fuzzy utility theory and FUF (Kirkwood, 1991) [18] were”

Before Figure 9 there is a table with no number or caption…

(…)

Unacceptable English…

Table 14. Compared with the evaluation results of other relevant decision-making models”

It is not possible to properly review a text with poor English….

Instead of focusing in the essential, the reviewer is worried with everything else…

Conclusions:

The way the authors write is not adequate to a scientific text. Language needs clear improvement:

“The main contribution of this study is that it provides construction project owners with an

619 improved method for contractor selection.”

See that authors link methodology to practical implications…

“In terms of the sustainability of the construction life cycle,

620 this method employs three selection criteria (duration, cost, and quality) to evaluate the economic

621 utility of bid commitment for the committee members or owners, evaluates the risk preference and

622 implementation possibility of bid commitment, and comprehensively analyses utility and

623 implementation possibility. By using the proposed method, wrong decisions can be avoided and

624 the decision-making quality can be improved.”

It goes on “In addition to the utilities of duration, cost, and quality commitments for committee members,

626 the possibility of D/C/Q commitments for ensuring successful implementation is also a key factor

627 that committee members consider when selecting contractors for construction projects. The

628 implementation possibility evaluation method presented in this study that combines the MPR and

629 FPR can effectively evaluate duration, cost, and quality commitments during the construction work

630 bidding phase to quantify the possibility of successful construction.”

It goes back to methodology:

“This study employed the Bayes fuzzy prospect model to be evaluated in four technique phases

632 including integrating duration discount, cost discount, and quality assurance of a bid and

633 determining the probability by external environment information that these commitments can be

634 implemented.”

And again practical implications:

“A bidder’s overall prospect value was evaluated using the aforementioned method.

635 This approach can help committee members and owners to successfully evaluate the overall utility

636 and risks pertaining to candidate contractors.”

Faced to the extense data, where are the real findings of this text?

As in abstract, the structure should be:

“Brief contextualization

Brief methodology

Main findings

Practical implications

References (scarce for such a long text, and also considering that many relate mathematical data) need update: 1 citation from 2019 in a paper submitted 2020…

A final table appears at the end… ?

According to previous comments and in my perspective, the structure of the text must be entirely revised, so that it follows a logic sequence.

I find the text very difficult to be reviewed. It has to do with the lack of consistence during the “flow” of the manuscript, where references to contextualization are mixed with the references to “this study”

The systematic use if “lists” all over the text, break the continuity.

Involving a questionnaire, an ethical committee must be involved, as informed consent.

Where is information about that?

I believe authors would benefit from asking for help from a colleague used to the entire publication process.

My main difficulty is the lack of coherence in the entire text. I cannot review it properly, then.

Reviewer 3 Report

The paper analyzes actual theme. All chapters of the paper is well done instead one chapter. To publish paper without literature overview is not allowed. In this paper, there is no literature overview. Mentioned only two old sources about contracts in construction. Only few cited papers are written in this decade. Four publications published in 2011 and one – in 2019.  The authors creating difficult model, which is done successfully. Congratulations for the authors. However they must overview already made models, where applied MCDM methods. There are many simple models, which allow quickly and successful to solve such cases.

Paper has theoretical and practical value, so it should be published. However should be showed this model place between other already published models. Should be shown it advantages and disadvantages.

Please find below the list of papers where analyzed MCDM methods application for contractor selection. Also, please find below the list of papers published during the last five years. Maybe it will help to prepare an overview chapter.

Recently were published paper, which analyze MCDM and Bayes methods interfaces. The journal Symmetry acknowledged it as the best article during 2019 (Symmetry-Basel 10(6) 205, 2018). Authors should hard work in order to prepare good overview and prepare discussion chapter. Then the value of this paper will grow and it will be well reading and citing. Congratulations for the authors, who reached nice results.

List of publications:

civil engineering journal-tehran 4(5): 1074-1086, 2018 journal of construction engineering and management 145(2): 04018123, 2018 open geosciences 10(1): 661-677, 2018 journal of grey system 29(4): 49-60, 2017 JCEM 23(6): 824-835, 2017 Information 9(3): 68, 2018 TEDE 24(5): 1866-1884, 2018 IEEE transactions on engineering management 58(4): 602-673, 2011 construction & building technology 49: 425-447, 2013 automation in construction 45: 151-152, 2014 advances in civil engineering 75489035, 2017 economic research-ekonomska istrazivanja 31(1): 1666-1716, 2018 economic research-ekonomska istrazivanja 28(1): 510-571, 2015 ACME 19(4): 1056-1071, 2019 Energies 12(13): 2481, 2019 Sustainability 11(2): 424, 2019 international journal of civil engineering 6(6A): 695-714, 2018 international journal of construction management 19(6): 492-508, 2019 Sustainability 11(22): 6444, 2019 JCEM 25(3): 241-251, 2019 JCEM 24(2): 155-166, 2018 JCEM 24(3): 254-264, 2018 TEDE 22(2): 210-234, 2016 TEDE 25(2): 258-276, 2019

The paper analyzes actual theme. All chapters of the paper is well done instead one chapter. To publish paper without literature overview is not allowed. In this paper, there is no literature overview. Mentioned only two old sources about contracts in construction. Only few cited papers are written in this decade. Four publications published in 2011 and one – in 2019.  The authors creating difficult model, which is done successfully. Congratulations for the authors. However they must overview already made models, where applied MCDM methods. There are many simple models, which allow quickly and successful to solve such cases.

Paper has theoretical and practical value, so it should be published. However should be showed this model place between other already published models. Should be shown it advantages and disadvantages.

Please find below the list of papers where analyzed MCDM methods application for contractor selection. Also, please find below the list of papers published during the last five years. Maybe it will help to prepare an overview chapter.

Recently were published paper, which analyze MCDM and Bayes methods interfaces. The journal Symmetry acknowledged it as the best article during 2019 (Symmetry-Basel 10(6) 205, 2018). Authors should hard work in order to prepare good overview and prepare discussion chapter. Then the value of this paper will grow and it will be well reading and citing. Congratulations for the authors, who reached nice results.

List of publications:

civil engineering journal-tehran 4(5): 1074-1086, 2018 journal of construction engineering and management 145(2): 04018123, 2018 open geosciences 10(1): 661-677, 2018 journal of grey system 29(4): 49-60, 2017 JCEM 23(6): 824-835, 2017 Information 9(3): 68, 2018 TEDE 24(5): 1866-1884, 2018 IEEE transactions on engineering management 58(4): 602-673, 2011 construction & building technology 49: 425-447, 2013 automation in construction 45: 151-152, 2014 advances in civil engineering 75489035, 2017 economic research-ekonomska istrazivanja 31(1): 1666-1716, 2018 economic research-ekonomska istrazivanja 28(1): 510-571, 2015 ACME 19(4): 1056-1071, 2019 Energies 12(13): 2481, 2019 Sustainability 11(2): 424, 2019 international journal of civil engineering 6(6A): 695-714, 2018 international journal of construction management 19(6): 492-508, 2019 Sustainability 11(22): 6444, 2019 JCEM 25(3): 241-251, 2019 JCEM 24(2): 155-166, 2018 JCEM 24(3): 254-264, 2018 TEDE 22(2): 210-234, 2016 TEDE 25(2): 258-276, 2019